# Machine learning multi-omics analysis reveals cancer driver dysregulation in pan-cancer cell lines compared to primary tumors

Lauren M. Sanders [1,2 ✉], Rahul Chandra[3], Navid Zebarjadi[2,4], Holly C. Beale [2,4], A. Geoffrey Lyle [2,4], Analiz Rodriguez[5], Ellen Towle Kephart [2], Jacob Pfeil[1,2], Allison Cheney [2,4], Katrina Learned[1,2], Rob Currie [1,2], Leonid Gitlin[6], David Vengerov[7], David Haussler [1,2,9], Sofie R. Salama [1,8,9] & Olena M. Vaske [2,4,9 ✉]

Cancer cell lines have been widely used for decades to study biological processes driving cancer development, and to identify biomarkers of response to therapeutic agents. Advances in genomic sequencing have made possible large-scale genomic characterizations of collections of cancer cell lines and primary tumors, such as the Cancer Cell Line Encyclopedia (CCLE) and The Cancer Genome Atlas (TCGA). These studies allow for the first time a comprehensive evaluation of the comparability of cancer cell lines and primary tumors on the genomic and proteomic level. Here we employ bulk mRNA and micro-RNA sequencing data from thousands of samples in CCLE and TCGA, and proteomic data from partner studies in the MD Anderson Cell Line Project (MCLP) and The Cancer Proteome Atlas (TCPA), to characterize the extent to which cancer cell lines recapitulate tumors. We identify dysregulation of a long non-coding RNA and microRNA regulatory network in cancer cell lines, associated with differential expression between cell lines and primary tumors in four key cancer driver pathways: KRAS signaling, NFKB signaling, IL2/STAT5 signaling and TP53 signaling. Our results emphasize the necessity for careful interpretation of cancer cell line experiments, particularly with respect to therapeutic treatments targeting these important cancer pathways.

[1] Department of Biomolecular Engineering, UC Santa Cruz, Santa Cruz, CA, USA. [2] UC Santa Cruz Genomics Institute, Santa Cruz, CA, USA. [3] Paul G. Allen School of Computer Science and Engineering, University of Washington, Seattle, WA, USA. [4] Department of Molecular, Cell and Developmental Biology, UC Santa Cruz, Santa Cruz, CA, USA. [5] Department of Neurosurgery, University of Arkansas for Medical Sciences, Little Rock, AR, USA. [6] Department of Microbiology and Immunology, University of California, San Francisco, San Francisco, California, USA. [7] Oracle Labs, Oracle Corporation, Pleasanton, CA, USA. [8] Howard Hughes Medical Institute, UC Santa Cruz, Santa Cruz, CA, USA. [9] These authors jointly supervised this work: David Haussler, Sofie R. Salama, Olena M. Vaske. ✉email: lmsh@ucsc.edu; olena@ucsc.edu

Tumor-derived cell lines provide a robust model environment for testing treatment hypotheses, identifying biomarkers of response to therapies, and studying underlying cancer biology. Cancer cell line models grow quickly, are comparatively cost-effective, and are readily available. Their integration into preclinical research has led to remarkable advances in cancer characterization and treatment[1].

However, additional genomic characterization of cancer cell lines has indicated that the transition from in vivo to in vitro may introduce key genomic alterations. One of the first groups to compare tumor and cell line used microarray gene expression profiles to identify breast cancer cell lines that seemed to be genetically inappropriate models for breast carcinoma[2]. As mutation detection has become more accurate, multiple studies have reported that head and neck cancer cell lines tend to harbor more mutations than their tumors of origin[3,4]. A recent study showed that colorectal cancer cell lines recapitulate colorectal tumor subtypes, but that cell lines have more mutations than tumors[5]. In general, the current consensus concerning cancer cell lines as primary tumor models is that cell lines share many of the original tumor characteristics, but can harbor genetic changes of poorly characterized significance; and that some cancer cell lines may not even be molecularly appropriate or representative models for their tumor of origin[6].

Nevertheless, cancer cell lines continue to be widely used in cancer research and therapeutic discovery. As the focus on molecularly targeted therapeutics grows, so does the need to thoroughly characterize how cancer cell lines diverge phenotypically from tumors due to their in vitro growth environment[6]. It is essential that preclinical researchers know which biological pathways behave similarly in vivo and in vitro, and even more importantly, which pathways demonstrate altered activity as a result of alterations in environmental signals and stressors in the in vitro growth setting. If a key pathway behaves differently in cell lines as compared to primary tumors, preclinical testing of a drug targeting that pathway will not accurately predict patient tumor response.

In order to characterize the specific pathway alterations that occur between primary tumors and tumor-derived cell lines, we analyze three types of high-throughput molecular data from The Cancer Genome Atlas (TCGA) and the Cancer Cell Line Encyclopedia (CCLE). We perform transcriptomic analysis of bulk RNA sequencing of TCGA and CCLE samples and identify a set of differentially expressed genes. We integrate micro-RNA sequencing from the same projects and identify an interaction network of micro-RNA (miRNA), long non-coding (lncRNA) and protein-coding genes that is aberrantly expressed in cell lines compared to tumors. This network implicates four key cancer driver pathways that are often the subject of preclinical drug evaluation in cell lines, but whose activity in cell lines is not representative of original tumors. We use proteomic quantification data from the same studies to demonstrate that the aberrant cancer driver pathway expression observed in cell lines extends to the proteomic level. We also demonstrate similar findings in separate datasets of single-cell RNA sequencing from tumors and cancer cell cultures.

## Results

### Support vector machine classifier identifies a set of genes differentially expressed between primary tumors and tumor-derived cell lines.

We hypothesized that genes with differential expression between the TCGA and CCLE datasets would represent differences in biological pathway activity. In order to identify novel sources of variation within these datasets, we eliminated immune-related genes because it is already known that cancer cell lines are unable to recapitulate the immune signatures of the primary tumors[7] (see "Methods", Supplementary Fig. 1 and Supplementary Data 1, Tab 2).

We then used a support vector machine (SVM) linear classifier within the Python *sklearn* module to identify genes (features) which are the most useful and important for classifying a new tumor or cell line based on a trained model[8]. Feature identification using SVM allowed us to identify the set of genes that best differentiate between tumor and cell line regardless of whether each gene is more highly expressed in the tumor group or the cell line group. The SVM approach has been shown to eliminate noisy results in a high-dimensional gene expression comparison, by drawing out the greatest sources of variation across all samples[9,10]. In order to ensure robust results, we repeated the classification on fifty different random 80/20 test/training splits of the data. After sorting the genes by their SVM-assigned feature importance coefficients, we merged the top 10% of genes from all fifty classifications, resulting in 1854 genes that were in the top 10% of most important genes for each classification (Supplementary Data 1, Tab 1). These genes included 54 lncRNA and 1799 protein-coding genes.

In order to characterize the functional significance of our SVM-derived gene set, we performed gene set enrichment analysis (GSEA) on the 1799 protein-coding genes using the Hallmark cancer pathway set from the Molecular Signatures Database (mSigDB v7.0)[11]. We found 27 gene sets with significant enrichment in the SVM-derived differentially expressed protein-coding genes (Fig. 1a and Supplementary Data 1, Tab 3).

Since lncRNA play known regulatory roles in normal tissue and in cancer, we hypothesized that the 54 differentially expressed lncRNA may be involved in regulating the differentially expressed coding genes, and may compose an interaction network with aberrant expression in cell culture. In order to characterize the functional interactions of these lncRNA, we employed miRNet, a tool that integrates multiple interaction databases for identification of lncRNA-miRNA and miRNA-gene regulatory networks[12]. miRNet identified 227 miRNA with known interactions to the 54 lncRNA. In turn, these 227 miRNA had 580 known gene targets among the 1799 differentially expressed coding genes ($P$ value $< 7.8^{-27}$, hypergeometric test). GSEA of the 580 coding genes revealed 24 Hallmark gene sets with significant enrichment ($P$ value $< 0.05$, Fig. 1b and Supplementary Data 1, Tab 3). Strikingly, 20 of these Hallmark pathways overlapped with the enriched pathways from the coding genes GSEA (Fig. 1c and Supplementary Data 1, Tab 3), indicating that the set of SVM-derived important lncRNA is closely involved in many of the same pathways as the set of SVM-derived important coding genes.

We categorized the pathways into 6 categories (cellular response, development, cancer driver, metabolism, blood, and immune). Notably, our results (Supplementary Data 1, Tab 3) recapitulate the findings of a recent study by Yu and colleagues which found key differences in developmental, cell cycle, and immune pathways between cell lines and tumors[7]. In this study, we were particularly interested in the five cancer driver pathways which overlapped between the coding genes GSEA and the lncRNA-derived GSEA, as these molecular pathways are most likely to be the focus of preclinical trials in cancer cell lines.

### Four main types of cancer driver pathways exhibit differential expression and protein levels in cancer cell lines compared to primary tumors.

The five cancer driver pathways with significant enrichment in both SVM-derived coding genes and lncRNA-related genes are KRAS Signaling Up and Down, P53 pathway,

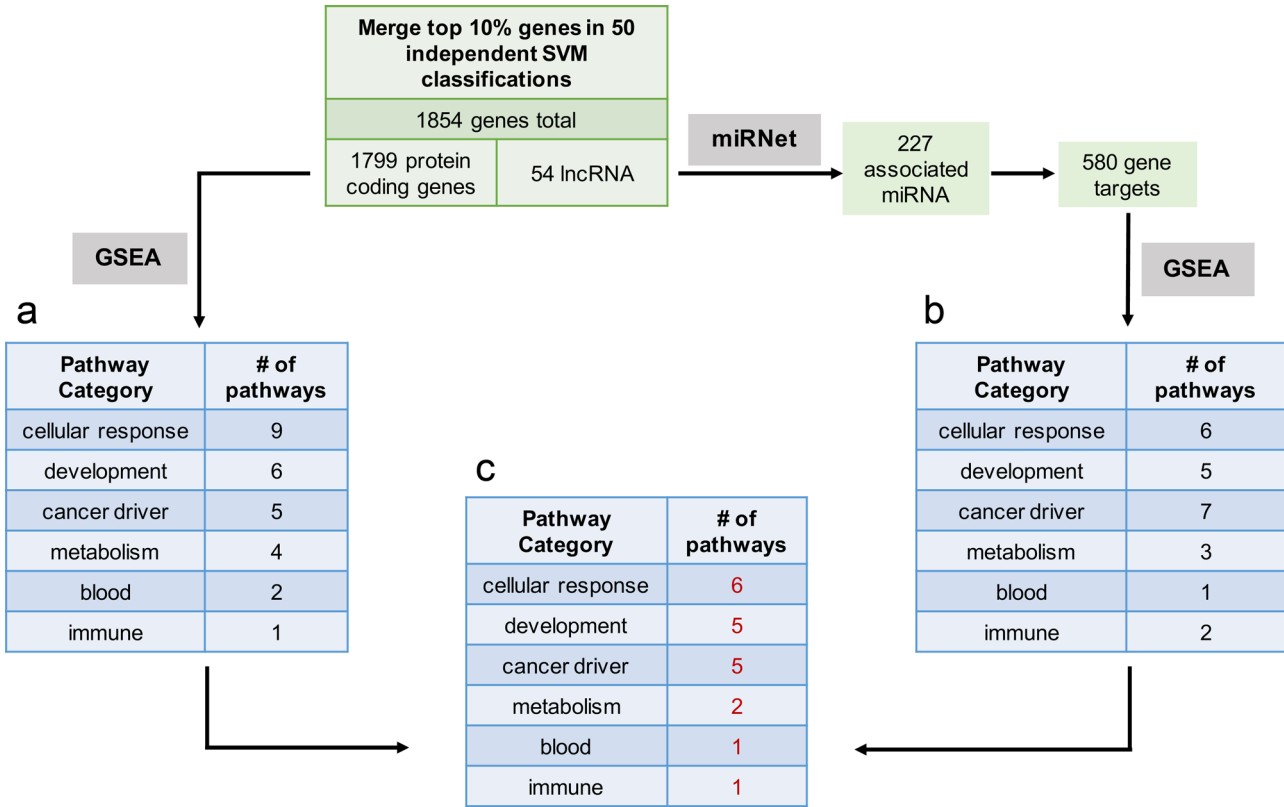

**Fig. 1 Workflow for support vector machine (SVM) classification of samples from TCGA and CCLE to assign feature importance to all genes. a** GSEA pathway enrichment results of 1799 protein-coding genes in the overlap of the top 10% most important genes in 50 independent SVM classifications. **b** GSEA pathway enrichment results of 580 protein-coding genes linked by miRNet databases to the 54 lncRNAs found in the overlap of the top 10% most important genes in 50 independent SVM classifications. **c** Overlap of pathways from the protein-coding gene GSEA and the lncRNA-based GSEA.

IL2/STAT5 signaling, and TNFA signaling via NFKB (Fig. 2a). With respect to KRAS signaling, since both pathways are a result of activated KRAS signaling, all subsequent analysis focuses on KRAS Signaling Up, which represents genes upregulated as a result of activated KRAS signaling. We focused on upregulated genes since most therapeutic approaches focus on reducing the activity of targeted genes[13].

Interestingly, all four pathways show much higher overall expression in tumors than in cell lines (Fig. 2b). As a control, we also examined the gene expression of the Hallmark PI3K-AKT-mTOR pathway, a cancer driver pathway that was not significantly enriched in SVM-derived genes ($P$ value < 0.426). This pathway did not show differential expression between CCLE and TCGA (Supplementary Fig. 2). We also verified that this signal was not a disease-specific artifact by repeating the SVM after subsetting the data to the disease with the largest number of samples in TCGA (BRCA) and the smallest number of samples (DLBC). The same four cancer driver pathways were identified in both analyses (Supplementary Data 1, Tab 4). These four pathways overall have similar correlation between tumor and cell line samples when evaluated on a disease-specific basis (Fig. 2d). We also verified that this signal is not related to tumor purity by performing ESTIMATE[14] tumor purity measurements on all TCGA samples and repeating the SVM comparison on the solid tumor types with the highest (KIRC) and lowest (PRAD) ESTIMATE scores (Supplementary Fig. 3). The four cancer driver pathways were identified as differentially expressed in both analyses (Supplementary Data 1, Tab 5).

We next examined whether the downregulation of these pathways extended beyond gene expression into protein activity. Proteomics quantification of many CCLE and TCGA samples was performed using Reverse Phase Protein Array (RPPA) in the MD Anderson Cell Lines Project (MCLP) and The Cancer Proteome Atlas (TCPA)[15,16]. Because the RPPA data include fewer than 250 proteins, we identified through literature review proteins that are normally highly expressed downstream of each cancer driver pathway, and examined their levels in the RPPA Level 4 Normalized cell line and tumor data (Fig. 2c). PIK3R1 (antibody PI3KP85) is activated subsequent to KRAS signaling, and Cyclin D1 (antibody CYCLIND1) is activated downstream of the P53 pathway[17,18]. STAT5 (antibody STAT5ALPHA) represents the protein counterpart of the *STAT5* gene. The antibody NFKBP65_pS536 binds to phosphorylated p65, one of the two protein subunits of NFKB. Phosphorylation of p65 is one of several molecular mechanisms known to activate the NFKB pathway[19,20].

We noted significantly lower protein expression of PIK3R1, Cyclin D1, and STAT5 in the cell-line data, consistent with our gene expression results in the corresponding pathways. This carries important implications for the applicability of preclinical drug tests against these targets in cancer cell lines. Interestingly, the phosphorylation level of p65 is higher in cell lines than tumors, opposite the gene expression of the NFKB signaling pathway. This suggests that p65 phosphorylation may be playing a different role in cell lines, and underscores the importance of examining multiple types of data to elucidate complex molecular interactions.

Because activation of the KRAS and TP53 pathways is associated with mutations in the *KRAS* and *TP53* genes[21,22], we investigated whether there is a correlation between diseases with heavy mutation burden in these genes and diseases with higher correlation between tumor and cell line, with the assumption that

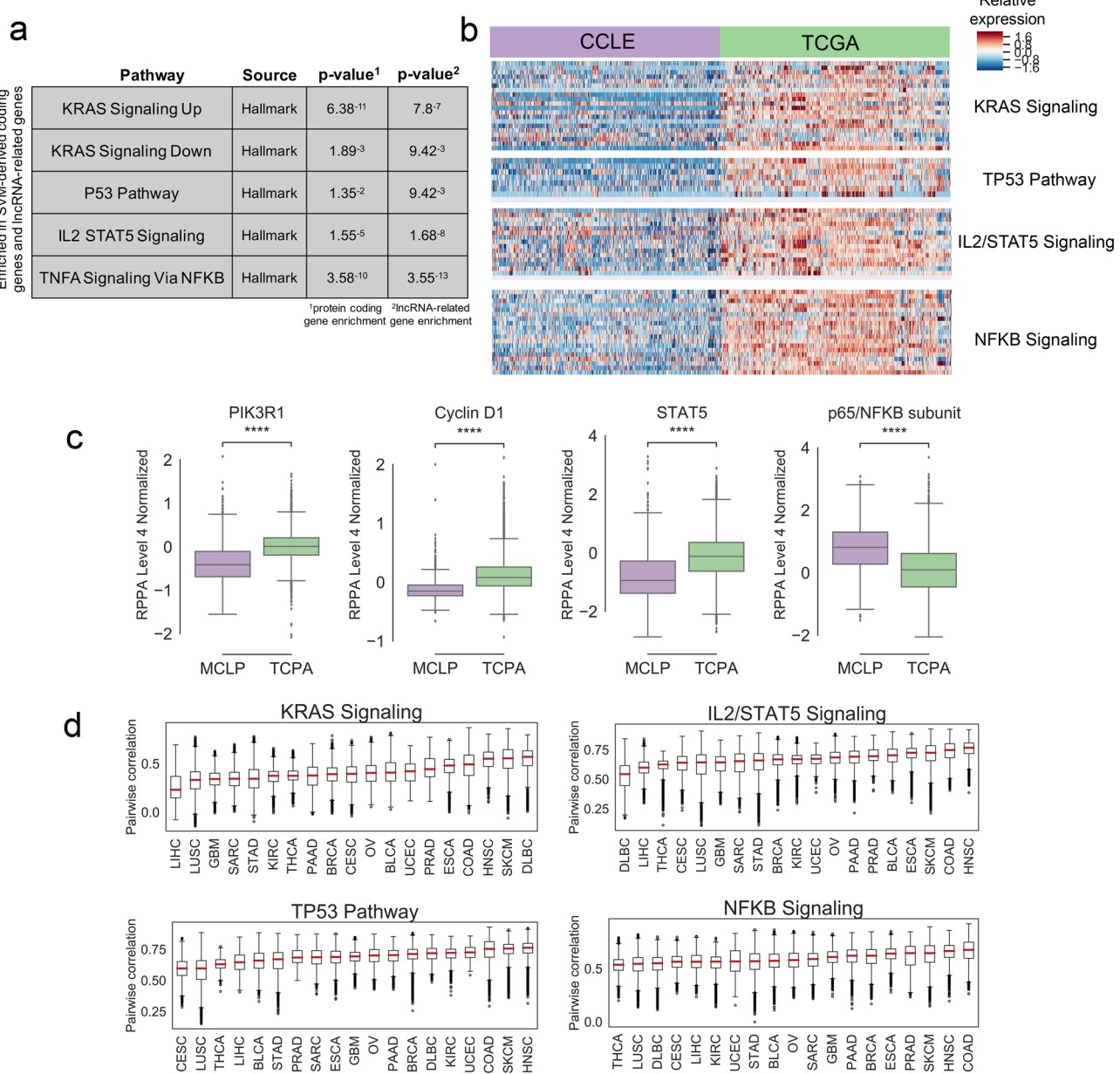

**Fig. 2 KRAS signaling, TP53 pathway, IL2/STAT5 signaling, and NFKB signaling are significantly enriched for genes with reduced expression in CCLE compared to TCGA. a** GSEA results for the cancer driver pathways which overlap with SVM-derived genes. *P* values are shown for the significance of gene overlap with SVM-derived protein-coding genes, and for genes linked by miRNet to SVM-derived lncRNA. **b** Heatmaps showing expression of SVM-identified genes in four cancer driver pathways in TCGA compared to CCLE. The samples shown are a random subset with equal representation from each dataset in each disease. **c** Boxplots showing overall protein quantification of representative proteins from each of the four cancer driver pathways in TCPA ($n = 5607$) and MCLP ($n = 646$) datasets (two-sided Mann–Whitney significance test; *$P$ value < 0.05, **$P$ value < 0.01, ***$P$ value < 0.001). **d** Boxplots show pairwise Spearman correlation scores between all CCLE and TCGA RNA-seq samples in each disease type, for all four cancer driver pathways. Plots are sorted by mean correlation.

cell lines derived from mutated tumors maintain those mutations. We found that there is no correlation between mutational burden and tumor-cell-line correlation; in fact, in the case of the TP53 pathway, there is a slight inverse correlation between the two factors (Supplementary Fig. 4). To further investigate the effect of *KRAS* and *TP53* mutation status on the differentially expressed cancer driver pathways, we subset the TCGA and CCLE datasets to samples carrying a non-silent *KRAS* or *TP53* mutation and repeated the SVM analysis (see "Methods"). In both the *KRAS*-mutant and *TP53*-mutant analysis, all four cancer driver pathways were again identified as differentially expressed in tumor compared to cell line. These results indicate that dysregulation of

these pathways occurs in cancer cell lines regardless of the mutational status of the primary tumor, and is unrelated to the activating DNA mutations.

**Dysregulation of a lncRNA-miRNA regulatory network in cancer cell lines is potentially associated with underexpression of key cancer pathways.** Because the four cancer driver pathways were derived in the context of lncRNA-related gene expression, we hypothesized that cell-line-specific dysregulation of lncRNA-based regulation programs may be associated with aberrant pathway-level gene expression. lncRNAs control gene expression

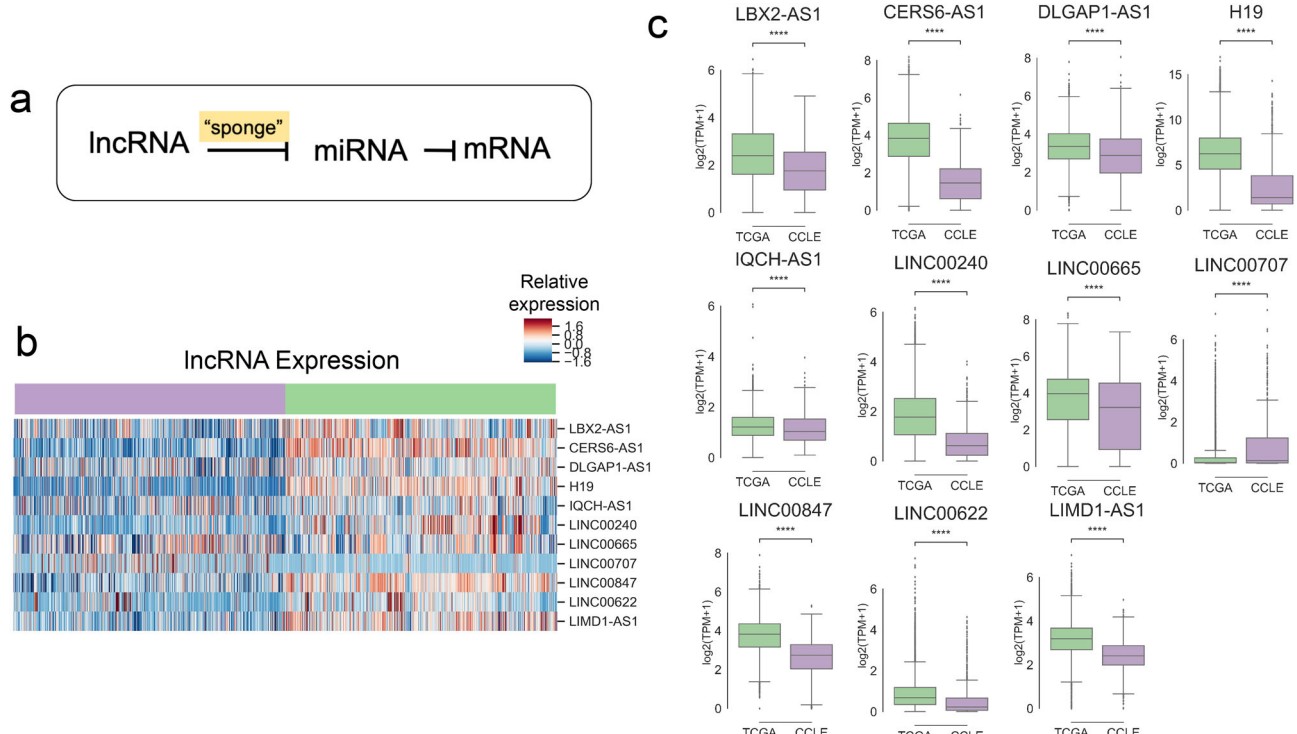

**Fig. 3 Long non-coding RNA associated with four cancer driver pathways are significantly underexpressed in cancer cell lines. a** In the "sponge" model of lncRNA gene expression regulation, lncRNA competitively inhibit miRNA which would otherwise be responsible for inhibiting mRNA. **b** Heatmap showing expression of 11 lncRNAs with miRNA-dependent associations to protein-coding genes in the four cancer driver pathways. The samples shown are a random subset with equal representation from each dataset in each disease. **c** Boxplots showing expression of the 11 lncRNA associated with four cancer driver pathways. All samples from both datasets are shown. (Two-sided Mann–Whitney significance test; *$P$ value < 0.05, **$P$ value < 0.01, ***$P$ value < 0.001).

in a tissue-specific manner, and one of their key regulatory mechanisms is by sequestering or "sponging" miRNA through base pairing interactions[23–25]. miRNA directly affect gene expression by binding mRNA and targeting them for degradation[26–28]. In this method of expression control, lncRNA regulate miRNA, while miRNA regulate gene expression (Fig. 3a).

To investigate potential non-coding RNA dysregulation in cell lines as compared to tumors, we focused on the 54 lncRNA identified as differentially expressed through the SVM classification (Fig. 1 and Supplementary Data 1, Tab 1). We used miRNet databases to link the 54 differentially expressed lncRNA to the 4 differentially expressed cancer driver pathways via shared miRNA interactions (Supplementary Data 2). Via miRNet, we found that 77 miRNA have known interactions both with genes in the four cancer driver pathways, and with 11 of the differentially expressed lncRNA (11 lncRNA: *LBX2-AS1, CERS6-AS1, DLGAP1-AS1, H19, IQCH-AS1, LINC00240, LINC00665, LINC00707, LINC00847, LINC00622, LIMD1-AS1*). With the exception of *LINC00707*, 10 of the 11 lncRNA are significantly underexpressed in CCLE (Fig. 3b, c). We hypothesized that the reduced cell-line expression of these lncRNA may be associated with expression changes in the downstream miRNA regulatory network, which in turn may be associated with aberrant expression of the four cancer driver pathways being controlled by the miRNA network.

In order to investigate this hypothesis, we leveraged publicly available miRNA sequencing (miRNAseq) data from CCLE and TCGA. We used the ComBat method to correct for experimental batch effects (see "Methods" and Supplementary Fig. 5)[29]. Sixty-nine of the 77 miRNA were quantified in both miRNAseq datasets, so we used these miRNA for all downstream

analyses (Supplementary Data 4, Tab 1). We calculated the log fold change (LFC) in expression between CCLE and TCGA for these 69 miRNA. Notably, over half of the miRNA ($n = 43$) are more highly expressed in cell lines than tumors. Cytoscape was used to visualize the lncRNA-miRNA-coding gene network colored by gene type or by LFC (Fig. 4a, b and Supplementary Data 4, Tab 2)[30].

In keeping with the lncRNA "sponge" regulatory model, the lncRNA are underexpressed in cancer cell lines, which could be involved with the observed overexpression of a majority of the miRNA whose expression is known to be kept in check by these lncRNA. The aberrant overexpression of inhibitory miRNA could explain the observed underexpression of key genes in four important cancer driver pathways in cancer cell lines. Consistent with this idea, we noted several miRNA with higher expression in cell lines that are known to play roles in regulation of the four cancer driver pathways. *mir-497, mir-195, mir-148a*, and *mir-152* directly inhibit genes in the KRAS/RAF/MEK/ERK pathway[31,32]. The TP53 pathway is repressed by *mir-339*, and the TP53-associated gene *TP53INP1* is regulated by *mir-92*[33,34]. *mir-519d* directly represses *STAT3*, a key gene in the IL2/STAT5 signaling pathway[35]. The NFKB pathway is activated by *mir-301a*, which has lower expression in cell lines compared to tumors, in keeping with lower NFKB activity in cell lines[36].

Because lncRNA and miRNA are known for cell-type-specific expression, we hypothesized that the observed dysregulation of lncRNA-miRNA expression networks is caused by biological selection for a subset of cancer cells which are more likely to survive the cell-line derivation process and thrive in a cell culture setting. Consistent with this hypothesis, both stem cell and epithelial cell-specific lncRNA and miRNA display reduced

expression in cancer cell lines. Specifically, CCLE samples have reduced expression of *H19*, a lncRNA strongly associated with the cancer stem cell state[37], but show increased expression of *mir-1* and *mir-206*, which promote cellular differentiation by blocking anti-differentiation signaling targets[38,39]. In addition, CCLE samples show reduced expression of *CERS6-AS1, IQCH-AS1,* and *LINC00240*, lncRNA implicated in mediating tight junctions or extracellular matrix interactions, which are features of epithelial and endothelial cells[40–43]. At the same time, CCLE samples have comparatively high expression of *mir-9*, which directly represses E-cadherin, a well-known epithelial marker[44]. E-cadherin repression is known to induce the epithelial-mesenchymal-transition, a process which plays a role in cancer progression from an epithelial state to a motile and invasive metastatic state[45]. CCLE samples display reduced expression of E-cadherin/*CDH1*, and higher expression of mesenchymal markers, including N-cadherin/*CDH2*, *MUC1*, and claudins *CLDN1, CLDN2, CLDN3*[46] (Supplementary Fig. 6).

The observed reduced epithelial and stem cell expression in cancer cell lines suggests that cancer cell culture conditions select for the subset of cancer cells with a mesenchymal, invasive and metastatic phenotype. Overall, these results indicate that selection against specific cancer cell types in tumor-derived cell lines may cause global downregulation of key cell-type-specific lncRNA, potentially allowing overexpression of a variety of miRNA, many of which play important roles in regulating cancer signaling pathways. However, more work is needed to fully investigate this hypothesis.

In light of recent research identifying a panel of 110 CCLE cell lines with the highest correlation to their primary tumor samples, the TCGA-110-CL[7], we examined whether these cell lines show more representative expression of the 4 cancer driver pathways. We repeated the SVM after subsetting the CCLE dataset to the TCGA-110-CL and the TCGA dataset to the tumor types in the TCGA-110-CL (Supplementary Data 1, Tab 6). Interestingly, the same four cancer driver pathways were again identified as differentially expressed (Supplementary Fig. 7). However, several metabolic, cellular response and developmental pathways that were identified in the original analysis were not identified here, including Hedgehog signaling, apical junction, and fatty acid metabolism (Supplementary Data 1, Tab 3). Overall, these results indicate that while the TCGA-110-CL cell-line panel is indeed more representative of its primary tumors by overall gene expression, our pathway-level examination reveals that caution must still be used when interpreting results involving targeting these four cancer driver pathways. This result is consistent with our hypothesis that the dysregulation of cancer driver signaling is driven by a loss of cellular heterogeneity overall in cancer cell lines.

**Single-cell RNA-seq analysis of hepatocellular carcinoma cell lines and tumor samples highlights that the differences in the expression of key cancer pathways are tumor-specific.** In order to further investigate the hypothesis that our results are driven by the selection of a specific malignant cell type in cancer cell line derivation, we leveraged previously published single-cell RNA sequencing (scRNAseq) data from hepatocellular carcinoma (HCC) cell lines and patient samples. Within the scRNAseq patient tumor data, we differentiated the malignant cells from the normal cell infiltrate (e.g., immune and stromal cells; Supplementary Data 3, Tab 1), in order to assess the impact of tumor purity on the observed cancer driver pathway dysregulation.

We used publicly available scRNAseq data from HCC cell lines HuH1 and HuH7[47] and from seven samples biopsied from two different HCC patients[48]. We assigned cell types based on the expression of published gene markers (Fig. 5a and Supplementary Data 3, Tab 1).

To see whether we could recapitulate the findings of our bulk RNA sequencing data analysis, we performed differential expression analysis between the cell line and tumor-cell populations. GSEA of the top 100 genes identified the KRAS, TP53, TNFA via NFKB, and IL6/STAT3 signaling pathways as enriched in genes differentially expressed in HCC tumor cells ($P$ value < 0.05), very similar to the bulk RNA sequencing analysis (Fig. 5b).

We then removed the non-malignant cells from the HCC tumor data and repeated the differential expression analysis and GSEA (Fig. 5c and Supplementary Data 3, Tab 2). We observed that the P53 Pathway and TNFA signaling via NFKB remained significantly enriched with $P$ value < 0.05, whereas the KRAS signaling and IL6/STAT3 signaling pathways were only significantly enriched with a less stringent $P$ value < 0.1. These results indicate that the observed overexpression of the TNFA signaling via NFKB and P53 Pathway in bulk RNA sequencing data from tumor samples is likely not related to tumor purity, whereas the observed KRAS signaling and IL2/STAT5 signaling overexpression may be in part attributable to lower tumor purity. We were unable to investigate the observed lncRNA and miRNA expression dysregulation due to the low sequencing depth of single-cell RNA sequencing data and low overall expression of non-coding RNA.

Finally, we leveraged the HCC scRNAseq data to investigate our hypothesis that cancer cell culture conditions select for the subset of cancer cells with a mesenchymal, invasive, and metastatic phenotype. We evaluated three well-known molecular subtypes of HCC tumors: S1 subtype is invasive and characterized by poor survival; S2 subtype tumors are larger, with poor survival; and S3 subtype is lower grade, with overall better survival[49]. These three subtypes are all represented in the seven scRNAseq HCC patient samples[48].

We wanted to see whether these three subtypes were also fully represented in the HCC cell-line data. We evaluated the expression of gene signatures associated with each subtype in the HCC cell line and malignant tumor cells, and observed higher overall expression of the S1 signature in cell lines, with higher expression of S2 and S3 signatures in the tumor samples (Fig. 5d). The enrichment of each signature in the top 100 differentially expressed genes in each sample is shown in Fig. 5e (Supplementary Data 3, Tab 2). The HCC cell-line population is only enriched for the most invasive, metastatic subtype S1, while the HCC tumor cells display variable enrichment for each subtype. These results are consistent with our hypothesis that cancer cell culture conditions select for the most invasive cell subtypes.

A limitation of the HCC analysis is that cancer and cell-line samples were not matched from the same patients. Therefore, we performed the same analysis using two sets of matched brain tumor and cell-line samples (melanoma brain metastases; Supplementary Figs. 8 and 9, respectively). We again observe enrichment of the same four cancer driver pathways, with the exception of the KRAS signaling pathway in the first analysis.

## Discussion
The ability to model and manipulate cancer cells has empowered therapeutic discovery since the derivation of the first cancer cell line[50]. Two-dimensional cancer cell cultures have enabled researchers to discover how cancers arise, characterize cancer cell types and growth patterns, and identify effective pharmaceuticals through drug screens[1]. Today, personalized tumor-derived 2D and 3D cultures are increasingly in use for the identification of precision therapies for individual patients[51–53]. The rise of

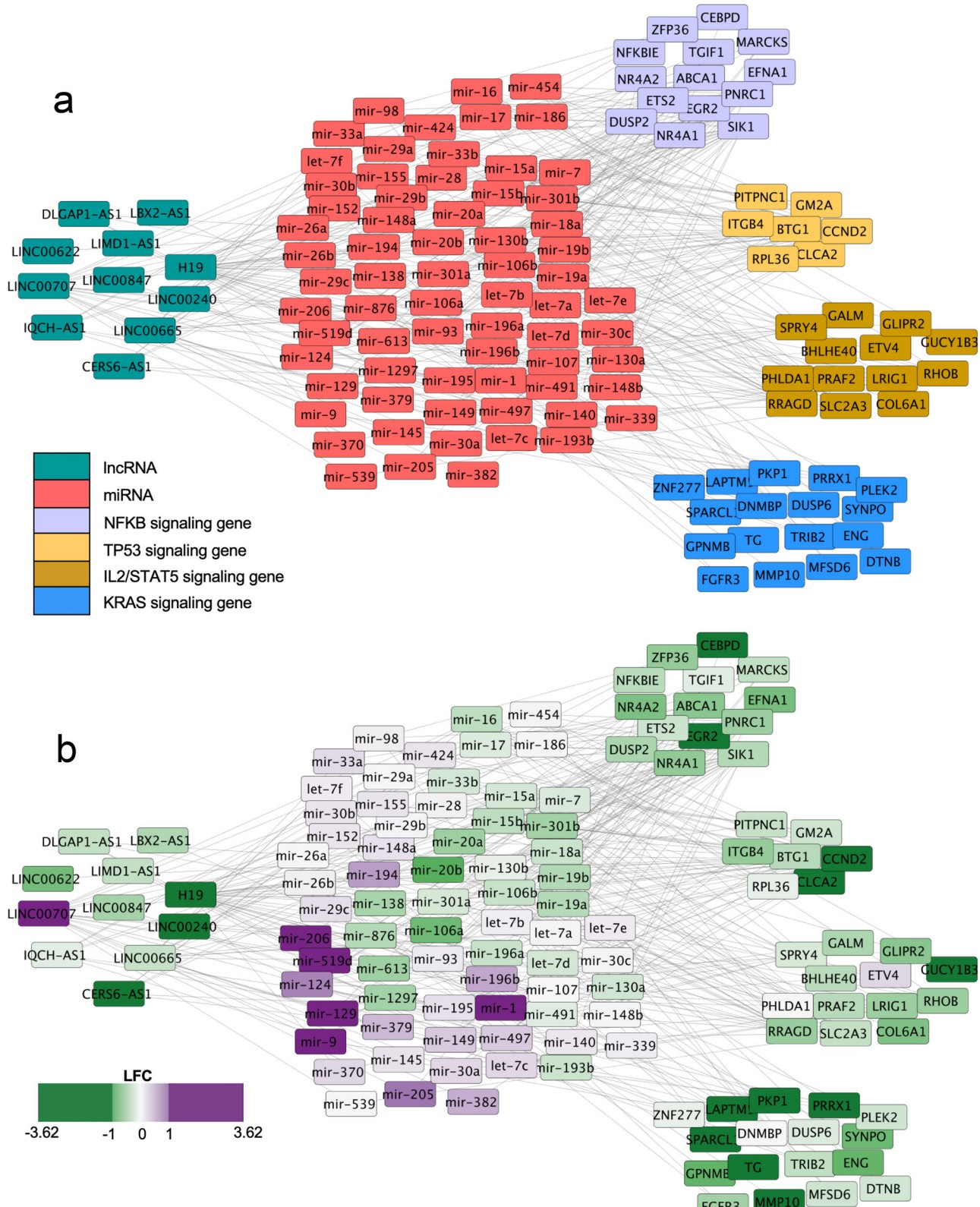

**Fig. 4 Dysregulation of lncRNA-miRNA regulatory network causes downregulation of key cancer driver pathways in tumor-derived cell lines. a** Types of genes are identified by color and positioning in the Cytoscape graph. Gene interactions from miRNet databases are denoted by gray lines. lncRNA are on the left, miRNA in the center, and differentially expressed protein-coding genes from each of the four cancer driver pathways are on the right side of the graph. **b** Positive LFC (purple) denotes higher expression in CCLE. Negative LFC (green) denotes higher expression in TCGA.

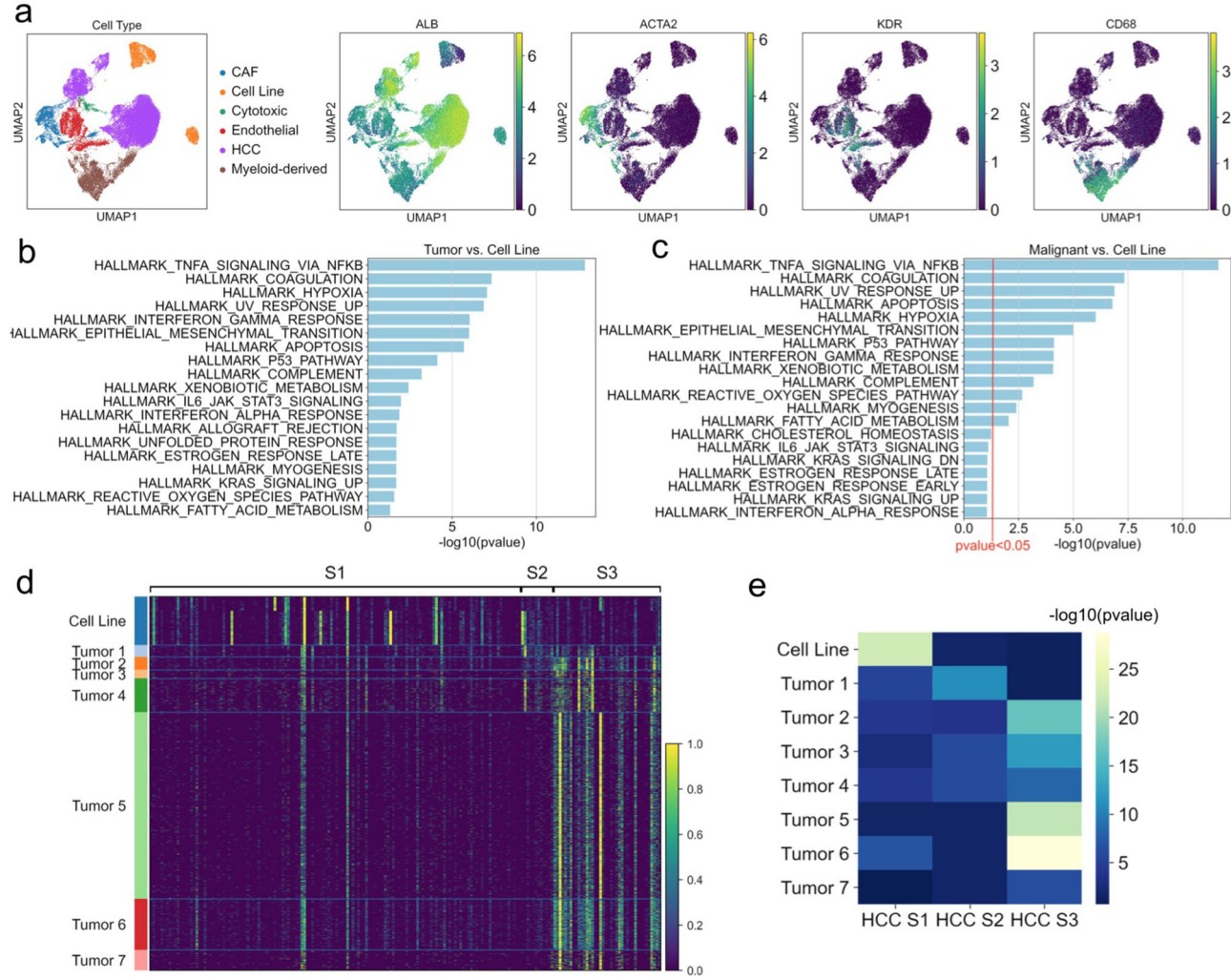

**Fig. 5 Single-cell RNA sequencing analysis of hepatocellular carcinoma tumor samples and cell lines. a** Single-cell RNA sequencing data clustering of hepatocellular carcinoma (HCC) cell line and tumor cells. **b** Gene set enrichment analysis of genes differentially overexpressed in all HCC tumor cells compared to HCC cell-line cells (*P* value < 0.05). **c** Gene set enrichment analysis of genes differentially overexpressed in only malignant HCC tumor cells compared to HCC cell-line cells (*P* value < 0.1, red line denotes *P* value < 0.05 cutoff). **d** Heatmap showing relative expression of genes from HCC molecular subtype gene signatures, in HCC cell line and tumor single-cell samples. **e** Heatmap showing −log10 *P* values for enrichment of differentially expressed genes in each HCC sample overlapping with the gene signatures for each HCC molecular subtype.

precision medicine and small-molecule inhibitor development brings an urgent need for molecular characterization of the cell culture models used widely for therapeutic development. Large-scale genomic efforts such as The Cancer Genome Atlas (TCGA) and the Cancer Cell Line Encyclopedia (CCLE) have enabled comprehensive comparison of cancer cell cultures and primary tumors.

Here, we provide a comparative multi-omic analysis of cancer cell lines and primary tumors by leveraging several types of genomic data from large public compendia. Our gene expression analysis reveals reduced expression of key cancer driver pathways, including KRAS signaling, TP53 pathway, IL2/STAT5 signaling, and NFKB signaling in cancer cell lines. These results are recapitulated in a comparative analysis of protein levels for key proteins from each pathway. Our analysis indicates the need for caution when interpreting in vitro preclinical testing results of inhibitors targeting these pathways.

In fact, the consequences of preclinical testing in cell lines against these pathways have already been felt. For example, during preclinical testing several MDM2/TP53 small-molecule inhibitors displayed potent cancer cell-line inhibitory activity, but

could only achieve partial tumor regression in xenograft models[54]. Subsequent optimization of these compounds led to success in clinical trials, but initial results in cell lines did not predict in vivo results. Targeting both wild-type and mutant KRAS in cancer has been notably unsuccessful; in particular, several high-throughput screens of KRAS-mutant cancer cell lines identified compounds which subsequently only partially reduced tumor volumes in xenograft models[55,56]. In contrast, targeting the cholesterol biosynthesis pathway, which is not differentially expressed in cancer cell lines by our analysis, has been promising both in vitro and in vivo through the usage of the statin family of drugs[57,58]. In addition, targeted inhibition of cyclin-dependent kinases (CDKs) by agents such as PD-0332991/palbociclib has been promising in phase I and II clinical trials, and this drug was originally identified in cancer cell lines[59,60]. CDKs are active during cell cycle entry from G0 and during the G2M checkpoint pathway, which were not differentially expressed in cancer cell lines by our analysis. We note that apart from the four cancer driver pathways identified as dysregulated in cell lines, preclinical testing in cell lines can in many cases reliably predict in vivo responses.

To identify potential causes of dysregulation of cancer driver signaling in cell lines, we analyzed the expression of lncRNA and miRNA implicated in regulation of these pathways. In cell lines, we found reduced expression of a set of lncRNA predicted to regulate a downstream network of regulatory miRNA, which are in turn overexpressed. Several of these miRNA are directly involved in specific inhibition of these cancer driver pathways, linking their overexpression to the observed reduced expression of cancer driver pathways.

We speculate that our results may indicate a partial loss of cancer stem cells in cancer cell culture due to the presence of serum in culture media. All CCLE cell lines were cultured in RPMI or DMEM media with 10% fetal bovine serum[61]. It is well known that the presence of serum in culture media encourages cellular differentiation, and cancer cell lines grown in serum-free conditions contain larger populations of cancer stem cells[62,63]. This hypothesis is supported by the markedly lower expression of stem cell-specific lncRNA in cell culture, and higher expression of pro-differentiation miRNA. Because cancer stem cell populations are known for their chemoresistance, and even small populations are thought to be capable of tumor recurrence[64–66], it is essential that preclinical models accurately model the response of cancer stem cells to potential therapeutics.

A recent study by Yu et al. also included a pan-cancer analysis of TCGA and CCLE data, finding that tumor type is not a large factor in overall tumor-cell-line correlation, but that some cell lines are poor models of their primary tumor type and tumor purity plays a large part in tumor-cell-line correlations[7]. Our study provides additional information by identifying the set of genes which accounts for the highest variation between human tumors and cell culture models, and by identifying specific, clinically important cancer driver pathways which are poorly recapitulated in cancer cell culture models. Additionally, our finding of global cancer driver pathway dysregulation in cell lines may help to explain biologically the previous study's finding of low correlation between some tumors and cell lines. We investigated the TCGA-110-CL cancer cell-line panel identified in the previous study, which has a higher overall cell-line-tumor gene expression correlation. The same four cancer driver pathways were identified by an SVM comparing these cell lines to disease-matched primary tumors, and in particular, the KRAS pathway had the lowest overall cell-line-tumor correlation. This indicates that even this subset of cancer cell lines does not fully recapitulate the cancer driver signaling of primary tumors, and studies on these pathways must be interpreted with caution.

In light of this finding that tumor purity impacts the correlation between tumor and cell line[7], we sought to assess the impact of tumor purity on our cancer driver pathway dysregulation findings by leveraging single-cell RNA sequencing data from HCC patients and cell lines. The TP53 and NFKB pathways remained significantly enriched in HCC tumor cells compared to cell lines when normal infiltrating cells were removed from the tumor-cell population, while the KRAS and IL6/STAT3 pathways were only significantly enriched with a less stringent statistical cutoff. These results indicate that the KRAS and IL2/STAT5 findings from the TCGA-CCLE gene expression comparison may be partially due to lower tumor purity. Moreover, this analysis indicates that normal cell infiltrate contributes a molecular signature to bulk gene expression data and highlights the utility of single-cell data. Finally, we found that the HCC cell lines most closely resemble the most invasive and metastatic HCC subtype, unlike the HCC tumor cells which resemble all three HCC subtypes, consistent with our hypothesis that cancer cell culture selects for the most invasive malignant cells. We saw similar results in an analysis of matched melanoma brain metastasis cell lines and tumors. A caveat of this analysis is that it focuses on only two types of solid tumor; a broader investigation of single-cell RNA sequencing data in this context from multiple tumor types and cell lines will be informative when those data become available.

Taken together, our results underscore the need for caution when interpreting preclinical cancer testing results in multiple model types, and point to specific signaling networks which can serve as litmus tests for the accuracy of past and future cancer laboratory models. We suggest several potential solutions to improve the efficacy of tumor-derived cell lines. Cancer cell culture in serum-free conditions may improve the maintenance of tumor stem cell populations and reverse the dysregulation of important regulatory gene networks. Specific efforts to model the immune microenvironment in cancer-derived organoids may improve cancer driver pathway expression related to the tumor microenvironment. A potential solution may be genetic manipulation of tumor-derived models with an emphasis on preserving or rescuing the intrinsic cancer driver pathway expression which is most at risk for dysregulation. Overall, this study provides much-needed genomics-based considerations for future preclinical cancer model development and result interpretation.

## Methods

**RNA sequencing data**. Gene expression transcripts per million (TPM) matrices from TCGA ($n$ samples = 10,535) and CCLE ($n$ samples = 933) were downloaded from the UCSC Xena browser. These data were processed uniformly through the TOIL UCSC RNA sequencing data processing pipeline (TCGA on v2.0.8; CCLE on v3.3.4) to remove technical batch effects[67]. Both datasets were normalized by log2(TPM + 1) and duplicate genes were averaged. Genes not expressed in 80% of samples were removed, and 20% of the lowest varying remaining genes were removed, leaving 46865 remaining genes. Both datasets were subset to the 19 overlapping cancer types for subsequent analysis (BRCA, LUSC, LIHC, DLBC, THCA, PRAD, OV, STAD, BLCA, KIRC, UCEC, COAD, SARC, CESC, SKCM, PAAD, HNSC, ESCA, GBM). All heatmaps use a random subset of samples from each dataset with equal numbers from each disease. The random subset method is used because the CCLE data holds approximately 1/10 the number of samples as the TCGA data, making it impossible to visually detect gene expression differences in heatmaps containing all samples from both datasets.

**Mutation data**. TCGA somatic mutation data were downloaded from UCSC Xena (xena.ucsc.edu). CCLE somatic mutation data were downloaded from the Broad Institute CCLE Database (portals.broadinstitute.org/ccle/data).

**micro-RNA sequencing data**. TCGA micro-RNA (miRNA) Illumina sequencing read counts data were downloaded from the Genomic Data Commons Data Portal[68]. CCLE Nanostring probe miRNA quantification data were downloaded from the Broad Institute CCLE database: https://portals.broadinstitute.org/ccle/data[69]. For dataset comparability, miRNA naming formats were harmonized, and duplicates were averaged. Because different miRNA sequencing methods were used in each dataset, ComBat was used to batch-correct the data[29]. Pre- and post-batch effect correction data were then log2(count + 1) normalized for downstream visualization and analysis. Supplementary Fig. 5 shows pre- and post-batch effect correction expression distributions of several housekeeping genes to validate successful correction[70,71].

**RPPA data**. Level 4 Reverse Phase Protein Array (RPPA) data for the TCGA and CCLE samples were downloaded from the The Cancer Proteome Atlas (TCPA) portal (https://tcpaportal.org/tcpa/download.html and http://tcpaportal.org/mclp/#/download). Both datasets were subset to the 16 overlapping cancer types for subsequent analysis (BLCA, BRCA, COAD, DLBC, HNSC, KIRC, LGG, LIHC, LUAD, OV, PAAD, PRAD, SARC, SKCM, STAD).

**Single-cell RNA sequencing data**. Hepatocellular carcinoma (HCC) circulating tumor cell and cell line 10x Genomics single-cell RNA sequencing datasets were downloaded from the Gene Expression Omnibus (accession GSE103867). The cell-line cells were selected based on the expression of *ALDH1A1*, based on the original study findings (see Fig. 4, Zheng et al.[23]). HCC tumor 10x Genomics single-cell RNA sequencing datasets from seven samples from two patients were downloaded from the Gene Expression Omnibus (accession GSE112271). All single-cell gene expression data analysis was performed using scanpy (v1.6.7) in Python (v3.6.8). The datasets were combined, and cells expressing >5000 genes and/or >30 mitochondrial genes were excluded, leaving 52,630 cells. Leiden clustering was performed using the top 40 Principal Components, cell types assigned based on cell-type markers (Supplementary Data 3, Tab 1), and the top 100 upregulated genes in the primary tumors were derived by a Wilcoxon rank-sum test. Primary patient-

derived melanoma brain metastasis tissue (CI0000035650 and CI0000035270) were obtained from the UAMS Tissue Biorepository and Procurement Core within a few hours of surgical removal where tumors were cultured and bio-banked. Consent was obtained from all patients prior to surgery (IRB# 228443). A board-certified neuropathologist confirmed all tumor histopathological diagnoses. Tumors were harvested intraoperatively with myriad device which mechanically dissociates the tumor (NICO corporation, Indianapolis, IN, USA). Tumor specimens were then digested with collagenase solution. Cells were then prepared for single-cell sequencing using the 10x Genomics platform (10x Genomics, Pleasanton, CA, USA) as previously described[72]. The remaining cells, following digestion, were cultured in Roswell Park Memorial Institute culture medium (RPMI) containing 10% fetal bovine serum (FBS) and 1% (v/v) antibiotic/antimitotic. Early passage cells (<P3) were grown to confluency and then harvested for single-cell RNA sequencing analysis as described above.

**Statistical gene selection via support vector machine**. In Python (v3.6.8), the sklearn module (v0.21.2) was used with linear kernel to train a support vector machine (SVM) on 50 random 80/20 splits of the merged TCGA-CCLE gene expression dataset, in which 50 different training sets (and corresponding test sets) were resampled from the same data. The top 10% of genes from each training run based on the magnitude of their feature weights coefficients were merged in a non-duplicate manner, to account for variation in prediction results based on the training and corresponding test set, resulting in 1858 genes. Gene set enrichment from this analysis revealed 26/100 enriched pathways were immune-related, so a non-redundant immune gene list was created by merging all genes from the 26 enriched immune pathways (Supplementary Data 1, Tab 1). A second SVM analysis was conducted on a set of 50 random 80/20 splits of the merged TCGA-CCLE gene expression dataset with the immune-related gene list removed, and the top 10% of genes from each run based on feature importance coefficients were merged, resulting in 1854 genes which were used in all downstream analysis. We verified that the cancer driver pathway signal identified in the second SVM analysis is not related to immune signaling by calculating pairwise correlation between SVM-identified cancer driver genes and the immune genes which were removed from the analysis (mean = 0.1, Supplementary Fig. 1).

**Statistics and reproducibility**. Expression comparisons of individual genes between datasets were performed using a two-sided Mann-Whitney significance test with significance defined as $P$ value < 0.05. Significantly enriched gene sets were identified using gene set enrichment analysis[11] with FDR $q$-value < 0.05. For bulk RNA-seq, heatmaps are generated using seaborn clustermap method in Python with z-scores calculated per gene. For single-cell RNA-seq, heatmaps are generated using scanpy heatmap method in Python with values scaled between 0 and 1. Figures were produced using pandas v0.25.3, matplotlib v3.0.3, and seaborn v0.9.0 in Python v3.6.8. Numerical data used to generate figures are available here: https://drive.google.com/drive/u/1/folders/16pqRgDIxS0Wj_H9rIwyjB11a0IE6TAXH All data were used for each figure, except when otherwise indicated.

**Reporting summary**. Further information on research design is available in the Nature Research Reporting Summary linked to this article.

## Data availability

The TCGA and CCLE processed gene expression data are publicly available at UCSC Xena (xena.ucsc.edu). The processed micro-RNA sequencing data are publicly available at the Genomic Data Commons and the Broad Institute CCLE database (portals.broadinstitute.org/ccle/data). The protein quantification data are publicly available at the The Cancer Proteome Atlas (TCPA) portal (tcpaportal.org/tcpa/download.html and tcpaportal.org/mclp/#/download). The mutation data are publicly available at UCSC Xena and the Broad Institute CCLE database. The hepatocellular carcinoma single-cell RNA sequencing data are publicly available at the following accessions: GSE103867 and GSE112271. The brain tumor single-cell RNA sequencing data are publicly available at the following Gene Expression Omnibus accession: GSE213519.

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

## Acknowledgements

This study was funded by American Association for Cancer Research NextGen Grant for Transformative Cancer Research Award (O.M.V.), St Baldrick's Foundation Consortium Award and Emily Beazley Kures for Kids Fund Hero Award (D.H.), Alex's Lemonade Stand Foundation for Childhood Cancer Research, Unravel Pediatric Cancer, Team G Childhood Cancer Foundation, the California Initiative to Advance Precision Medicine and Live for Others Foundation. D.H. is a Howard Hughes Medical Institute Investigator. O.M.V. holds a Colligan Presidential Chair in Pediatric Genomics. This study makes use of data generated by the St. Jude Children's Research Hospital—Washington University Pediatric Cancer Genome Project; the St. Jude Rhabdomyosarcoma data were part of SJC-DS-1001, which was obtained with permission from St. Jude Cloud (https://www.stjude.cloud)—a publicly accessible pediatric genomic data resource requiring approval for controlled data access: Chen et al.[73].

## Author contributions

L.M.S. contributed to conception, design, interpretation, and writing. R. Chandra, D.V., A.G.L., N.Z., and H.C.B. contributed data analysis. E.T.K., J.P., A.C., K.L., R. Currie, and A.R. contributed to data acquisition. L.G., D.H., S.R.S., and O.M.V. contributed interpretation and manuscript revision.

## Competing interests

The authors declare no competing interests.
