## [Peer Review File · Communications Biology]

Reviewers' comments:

Reviewer #1 (Remarks to the Author):

Thank you for the opportunity to review this provocative paper. The authors have taken on the important and significant task of measuring how accurately cancer cell line models recapitulate the gene expression profiles of primary tumors. They find that several key cancer-related pathways are consistently expressed at different levels between cell lines and tumors, and perform correlative studies on miRNAs and lncRNAs that suggest that these differences may be due to upstream changes in the lncRNA->miRNA->mRNA regulatory axis. These findings add another layer to the growing volume of evidence that cancer cell line models must be treated with considerable caution.

However, my enthusiasm is tempered by a confounding factor that the authors fail to address. Unlike cell lines, which are more-or-less pure cell type populations, tumours are organ-like and consist of multiple cell types including the cancer cell itself, normal epithelium, fibroblasts, endothelial cells, inflammatory cells and others. The cancer cell compartment varies considerably in the degree of its contribution to the total. Hence when one performs bulk transcriptome profiling, the transcriptional profile represents both tumour and non-tumour contributions, and this will vary depending on the mixture of cell types. Yu et al (Nature Communications 10:3574 2019 "Comprehensive transcriptomic analysis of cell lines as models of primary tumors across 22 tumor types", reference -7- in the manuscript) partially corrected for this confounder by stratifying the primary tumours by their overall purity, and observed that the correlation between primary and cell lines increased in higher cellularity tumors. In the case of the current paper, while immune signatures were removed, there was neither a correction for purity and cellular heterogeneity nor a discussion of this confounder.

I would suggest a number of potential improvements to the analysis:

1. Stratify by tumour purity as was done by Yu et al.
2. Apply CIBERSORT or a similar cell type expression deconvolution algorithm in order to reduce the signal from endothelial cells, fibroblasts, immune cells and other normal stromal components.
3. Try to reproduce the major findings using a single cell RNA-seq tumour data set, after clustering the cells and removing those clusters that correspond to stromal/normal cell types.

If none of these is practicable, the paper needs at least to acknowledge and discuss why the issue is unlikely to change the conclusions.

Reviewer #2 (Remarks to the Author):

Sanders and colleagues perform a pan-cancer comparison of tumor and cancer cell line RNA and protein expression data, focusing largely on pathway-level differences in mRNA expression, and a proposed relationship to differential lncRNA and miRNA expression. The question of systematic differences between cancer cell lines and tumors is critically important, and the proposed link to lncRNA and miRNA expression is intriguing; however I did not find the analysis strategy sufficiently detailed and rigorous (or clearly described) to support the authors' interpretations. I also thought that there was insufficient discussion of, or comparison to, related previous work.

Major points

1) The similar recent work by Yu et al. (Nat Commun., 2019) is not discussed in sufficient detail. The work by Yu et al. also looked at pan-cancer differences in cell line tumor expression differences at both the gene and pathway level. I found it odd that the authors cited the Yu et al. paper for the TCGA-110-CL panel, but did not seem to discuss the very related (but in some ways seemingly different) results.

2) I found the analysis used to identify differentially expressed genes across cancer types to be problematic in a few ways. Most importantly: I did not find the approach used to account for non-

cancer cell contamination in the tumor samples to be convincing. Simply removing immune related genes post-hoc may account for much of the difference, but expression of other genes are also influenced by the presence of stromal and immune cells. Supervised approaches to explicitly control for these biases (such as used in Yu et al.), or even additional post-hoc analysis to argue that non-tumor cell contamination is not a significant bias in their downstream analysis would be important. Additionally, I believe standard differential expression analysis would be a more direct and interpretable approach to address this question (as used in Yu et al.). Importantly, this approach would allow for controlling key covariates, such as disease type/subtype. As it stands, I believe the global classifier-based analysis could be biased by differences in the cancer type composition of the cell line and tumor datasets. (More minor) Other details of this analysis were confusing to me. For example, it wasn't clear why the authors used 5-fold cross-validation per se for analysis of feature importance in the model (rather than prediction accuracy). I think the reason was to ensure robustness of the results, but I believe bootstrap resampling would be more appropriate to assess this. Similarly, the decision to merge the top 10% most important genes across 50 such samples seems arbitrary.

2) An alternative explanation for differential expression in some of these pathways (KRAS and TP53 signaling) would be differences in the prevalence of genomic alterations in these pathways. Indeed, higher rates of mutations in TP53 have been shown (Iorio et al. Cell, 2016). The authors do acknowledge this possibility, but their analysis looking at overall mutation rates per disease type compared to within-pathway expression similarity was confusing/unconvincing to me. Instead, I believe more careful differential expression analysis, controlling for mutation status and CNAs of key cancer drivers in individual samples would be needed.

3) To my mind, the suggested lncRNA -> miRNA -> mRNA mechanism for the observed pathway dysregulation is the most intriguing aspect of the paper, but is not convincingly established. The authors' own speculation about cell type specific differences being a potential explanation raises the question of whether these differences could be due to differences in the underlying composition of the cell line and tumor datasets (i.e. by disease type, or subtype), or potentially even related to contamination from non-tumor cells. The comparisons in Supplementary Fig. 6 for example gloss over this important aspect.

The protein expression comparisons seemed cherry-picked to me. More unbiased analysis there would help support the findings from mRNA data (though presumably the same challenges of controlling for differences in disease type composition and contamination from non-cancer cells would need to be addressed).

4) In general I did not find that the methods were described in sufficient detail. C I found the correlation analysis difficult to interpret. Are the authors arguing that the pattern of expression within these particular pathways is specifically divergent between cell lines and tumors? Assuming they were computing correlation coefficients between cell line and tumor samples using only genes from these particular pathways, this is likely complicated by disease-specific differences in how variably expressed the genes in those pathways are. For instance, in some disease types there might be less variation in expression across genes within a given pathway, and hence lower correlations between any pair of samples (even within tumors).

Minor points

1) It wasn't clear to me why the authors chose to focus on the 'KRAS Signaling Up' pathway per se.

The argument in section B that differences in overall expression were not a disease-specific artifact was not convincing.

2) Why show a random subset of samples in the heatmap plots? These plots could include some additional annotations (such as disease type) or structure (such as hierarchical clustering). As it stands they didn't seem very informative.

3) Why use miRNet predictions to predict DE miRNA from lncRNA, rather than just using the miRNA data itself?

4) It's odd that all pathways seem to show higher expression in tumors than cell lines. I wonder if this could be an artifact of the relative nature of expression data and the elimination of immune or non-cancer expression data? In other words, the expression data might need to be renormalized by an adjusted library size after removal of the putative immune genes.

5) Proteomics data now exists for >300 cell lines and could be used to provide more systematic comparison of protein expression levels (Nusinow et al., Cell 2020).

6) Testing for differences in individual miRNAs after ComBat correction seems odd. I believe the only reason that these differences aren't all identically 0 is based on assumptions of the batch correction model (empirical Bayes moderation methods). I wasn't sure how to interpret this.

7) How do their analyses say anything about PDX and organoids "better recapitulating cell type heterogeneity and cellular interactions"?

Reviewer remarks are shown in **blue**

New additions to the manuscript are shown in **red**

Reviewer #1 (Remarks to the Author):

Thank you for the opportunity to review this provocative paper. The authors have taken on the important and significant task of measuring how accurately cancer cell line models recapitulate the gene expression profiles of primary tumors. They find that several key cancer-related pathways are consistently expressed at different levels between cell lines and tumors, and perform correlative studies on miRNAs and lncRNAs that suggest that these differences may be due to upstream changes in the lncRNA->miRNA->mRNA regulatory axis. These findings add another layer to the growing volume of evidence that cancer cell line models must be treated with considerable caution.

However, my enthusiasm is tempered by a confounding factor that the authors fail to address. Unlike cell lines, which are more-or-less pure cell type populations, tumours are organ-like and consist of multiple cell types including the cancer cell itself, normal epithelium, fibroblasts, endothelial cells, inflammatory cells and others. The cancer cell compartment varies considerably in the degree of its contribution to the total. Hence when one performs bulk transcriptome profiling, the transcriptional profile represents both tumour and non-tumour contributions, and this will vary depending on the mixture of cell types. Yu et al (Nature Communications 10:3574 2019 "Comprehensive transcriptomic analysis of cell lines as models of primary tumors across 22 tumor types", reference -7- in the manuscript) partially corrected for this confounder by stratifying the primary tumours by their overall purity, and observed that the correlation between primary and cell lines increased in higher cellularity tumors. In the case of the current paper, while immune signatures were removed, there was neither a correction for purity and cellular heterogeneity nor a discussion of this confounder.

I would suggest a number of potential improvements to the analysis:

1. Stratify by tumour purity as was done by Yu et al.
2. Apply CIBERSORT or a similar cell type expression deconvolution algorithm in order to reduce the signal from endothelial cells, fibroblasts, immune cells and other normal stromal components.
3. Try to reproduce the major findings using a single cell RNA-seq tumour data set, after clustering the cells and removing those clusters that correspond to stromal/normal cell types.

If none of these is practicable, the paper needs at least to acknowledge and discuss why the issue is unlikely to change the conclusions.

We thank the reviewer for the thoughtful comments and excellent suggestions. In order to assess the impact of tumor purity on our analysis, we performed two additional analyses as recommended by the reviewer.

First, we scored all TCGA tumor samples for tumor purity using ESTIMATE algorithm (Yoshihara et al., *Nature Communications* 2013). The results are visualized in Supplementary Fig. 3:

We then repeated the SVM comparative analysis after subsetting the TCGA and CCLE data to the tumor type with the highest overall tumor purity scores (KIRC) or the lowest overall scores (PRAD). The same four cancer driver pathways were identified as overexpressed in tumors in both analyses, indicating limited impact of tumor purity on our initial results. We describe this analysis on page 7, lines 137-141:

“We also verified that this signal is not related to tumor purity by performing ESTIMATE¹⁴ tumor purity measurements on all TCGA samples and repeating the SVM comparison on the solid tumor types with the highest (KIRC) and lowest (PRAD) ESTIMATE scores (Supplementary Fig. 3). The 4 cancer driver pathways were identified as differentially expressed in both analyses (Supplementary Table 1).”

Second, we leveraged two publicly available single cell RNA sequencing datasets from hepatocellular carcinoma (HCC) patient biopsies and tumor-derived cell lines. We performed differential expression analysis between the tumor and cell line HCC data, before and after removing the non-malignant cells from the tumor data (Supplementary Table 4). Before removing the normal infiltrate, we again identified the same 4 cancer driver pathways as overexpressed in the HCC tumor cells. However, after removing the normal infiltrate, only the TP53 pathway and the NFKB pathway were enriched in the HCC tumor cells with standard p value <0.05 . With less stringent p value <0.1 , the KRAS and IL2/STAT5 pathway were again enriched. This analysis indicates a potential partial impact of tumor purity on the KRAS and IL2/STAT5 results, at least in HCC tumor cells. We describe this analysis in section D, pages 17-19, lines 271-313.

Reviewer #2 (Remarks to the Author):

Sanders and colleagues perform a pan-cancer comparison of tumor and cancer cell line RNA and protein expression data, focusing largely on pathway-level differences in mRNA expression, and a proposed relationship to differential lncRNA and miRNA expression. The question of systematic differences between cancer cell lines and tumors is critically important, and the proposed link to lncRNA and miRNA expression is intriguing; however I did not find the analysis strategy sufficiently detailed and rigorous (or clearly described) to support the authors' interpretations. I also thought that there was insufficient discussion of, or comparison to, related previous work.

Major points

1) The similar recent work by Yu et al. (Nat Commun., 2019) is not discussed in sufficient detail. The work by Yu et al. also looked at pan-cancer differences in cell line tumor expression differences at both the gene and pathway level. I found it odd that the authors cited the Yu et al. paper for the TCGA-110-CL panel, but did not seem to discuss the very related (but in some ways seemingly different) results.

Thank you to the reviewer for the suggestion. We would like to highlight the original analysis we performed (pages 13-14, lines 255-269) in which we examined whether the TCGA-110-CL panel better recapitulated the cancer driver signaling that we had identified as differentially expressed between TCGA tumors and CCLE cell lines. We report the findings, which indicate that the TCGA-110-CL panel better recapitulates overall gene expression but still exhibits underexpression of the 4 cancer driver pathways we identified.

In order to address the reviewer's concern, we have included an extended, in-depth discussion of the differences between the Yu et al. study and our study in the Discussion (page 25, lines 418-434).

2) I found the analysis used to identify differentially expressed genes across cancer types to be problematic in a few ways.

Most importantly: I did not find the approach used to account for non-cancer cell contamination in the tumor samples to be convincing. Simply removing immune related genes post-hoc may account for much of the difference, but expression of other genes are also influenced by the presence of stromal and immune cells. Supervised approaches to explicitly control for these biases (such as used in Yu et al.), or even additional post-hoc analysis to argue that non-tumor cell contamination is not a significant bias in their downstream analysis would be important.

We thank the reviewer for raising a very important point and providing some excellent recommendations for additional analysis. In order to assess the impact of tumor purity on our analysis, we performed two additional analyses.

First, we scored all TCGA tumor samples for tumor purity using ESTIMATE algorithm (Yoshihara et al., *Nature Communications* 2013). The results are visualized in Supplementary Fig. 3:

We then repeated the SVM comparative analysis after subsetting the TCGA and CCLE data to the tumor type with the highest overall tumor purity scores (KIRC) or the lowest overall scores (PRAD). The same four cancer driver pathways were identified as overexpressed in tumors in both analyses, indicating limited impact of tumor purity on our initial results. We describe this analysis on page 7, lines 137-141:

“We also verified that this signal is not related to tumor purity by performing ESTIMATE¹⁴ tumor purity measurements on all TCGA samples and repeating the SVM comparison on the solid tumor types with the highest (KIRC) and lowest (PRAD) ESTIMATE scores (Supplementary Fig. 3). The 4 cancer driver pathways were identified as differentially expressed in both analyses (Supplementary Table 1).”

Second, we leveraged two publicly available single cell RNA sequencing datasets from hepatocellular carcinoma (HCC) patient biopsies and tumor-derived cell lines. We performed differential expression analysis between the tumor and cell line HCC data, before and after removing the non-malignant cells from the tumor data (Supplementary Table 4). Before removing the normal infiltrate, we again identified the same 4 cancer driver pathways as overexpressed in the HCC tumor cells. However, after removing the normal infiltrate, only the TP53 pathway and the NFKB pathway were enriched in the HCC tumor cells with standard p value <0.05 . With less stringent p value <0.1 , the KRAS and IL2/STAT5 pathway were again enriched. This analysis indicates a potential partial impact of tumor purity on the KRAS and IL2/STAT5 results, at least in HCC tumor cells. We describe this analysis in section D, pages 17-19, lines 271-313.

Also, we demonstrated that the remaining gene expression is not influenced by immune signaling (from the Methods, page 30, lines 542-545):

“We verified that the cancer driver pathway signal identified in the second SVM analysis is not related to immune signaling by calculating pairwise correlation between SVM-identified cancer driver genes and the immune genes which were removed from the analysis (mean=0.1, Supplementary Fig. 1).”

Additionally, I believe standard differential expression analysis would be a more direct and interpretable approach to address this question (as used in Yu et al.). Importantly, this approach would allow for controlling key covariates, such as disease type/subtype. As it stands, I believe the global classifier-based analysis could be biased by differences in the cancer type composition of the cell line and tumor datasets.

Thank you to the reviewer for the suggestion. Traditional differential expression analysis uses linear gene models to identify lists of genes with overall higher expression in one group of samples compared to the rest. This is unlike the SVM, which trains on the data to blindly identify each group on the basis of variables (genes). The SVM returns the variables/genes which are most important for separating the two groups. In our study, leveraging the SVM rather than differential expression analysis allows us to conduct a completely unsupervised analysis in which the genes returned account for the highest variation, regardless of whether they are more highly expressed in one group compared to the other. The SVM approach eliminates noisy results in a high-dimensional genomics comparison by drawing out the greatest sources of variation across all samples. We have clarified our choice of method in the manuscript on page 5, lines 87-93:

“We then used a support vector machine (SVM) linear classifier within the Python sklearn module to identify genes (features) which are the most useful and important for classifying a new tumor or cell line based on a trained model⁸. Unsupervised feature identification using SVM allowed us to identify the set of genes which best differentiate between tumor and cell line regardless of whether each gene is more highly expressed in the tumor group or the cell line group. The SVM approach has been shown to eliminate noisy results in a high-dimensional gene expression comparison, by drawing out the greatest sources of variation across all samples^{9,10}.”

As we were interested in identifying a global cell line signature we avoided stratifying by disease type. To ensure that our results were not disease-specific we included an analysis of BRCA and DLBC (the largest and smallest diseases in TCGA), detailed in the manuscript on page 7, lines 137-141:

“We also verified that this signal was not a disease-specific artifact by repeating the SVM after subsetting the data to the disease with the largest number of samples in TCGA (BRCA) and the smallest number of samples (DLBC). The same 4 cancer driver pathways were identified in both analyses (Supplementary Table 1).”

(More minor) Other details of this analysis were confusing to me. For example, it wasn't clear why the authors used 5-fold cross-validation per se for analysis of feature importance in the model (rather than prediction accuracy). I think the reason was to ensure robustness of the results, but I believe bootstrap resampling would be more appropriate to assess this. Similarly, the decision to merge the top 10% most important genes across 50 such samples seems arbitrary.

We thank the reviewer for identifying confusing aspects of our analysis. We have clarified our approach in the Methods on page 29, lines 531-536: *“In Python (v3.6.8), the sklearn module*

(v0.21.2) was used with linear kernel to train a support vector machine (SVM) on 50 random 80/20 splits of the merged TCGA-CCLE gene expression dataset, *in which 50 different training sets (and corresponding test sets) were resampled from the same data. The top 10% of genes from each training run based on their feature weights coefficients were merged in a non-duplicate manner, to account for variation in prediction results based on the training and corresponding test set, resulting in 1858 genes.*”

We leveraged bootstrap resampling by performing 50 random 80/20 splits in which we resampled 50 different training sets (and corresponding test sets) from the same data. We merged the top 10% of most important genes from the 50 bootstrap resamples of the data because we found the prediction results to vary based on the particular training (and correspondingly test) set that was resampled, and we didn't want to present conclusions based on a single random sample.

2) An alternative explanation for differential expression in some of these pathways (KRAS and TP53 signaling) would be differences in the prevalence of genomic alterations in these pathways. Indeed, higher rates of mutations in TP53 have been shown (Iorio et al. Cell, 2016). The authors do acknowledge this possibility, but their analysis looking at overall mutation rates per disease type compared to within-pathway expression similarity was confusing/unconvincing to me. Instead, I believe more careful differential expression analysis, controlling for mutation status and CNAs of key cancer drivers in individual samples would be needed.

Thank you to the reviewer for an excellent recommendation. Per the reviewer's comment, we subset the TCGA and CCLE samples to only KRAS-mutant or only TP53-mutant samples and performed our SVM differential expression analysis between tumor and cell line within each mutation group. We describe the results on page 9, lines 181-186 of the manuscript:

“To further investigate the effect of KRAS and TP53 mutation status on the differentially expressed cancer driver pathways, we subset the TCGA and CCLE datasets to samples carrying a non-silent KRAS or TP53 mutation and repeated the SVM analysis (see Methods). In both the KRAS-mutant and TP53-mutant analysis, all four cancer driver pathways were again identified as differentially expressed in tumor compared to cell line (Supplementary Table 1).”

3) To my mind, the suggested lncRNA -> miRNA -> mRNA mechanism for the observed pathway dysregulation is the most intriguing aspect of the paper, but is not convincingly established. The authors' own speculation about cell type specific differences being a potential explanation raises the question of whether these differences could be due to differences in the underlying composition of the cell line and tumor datasets (i.e. by disease type, or subtype), or potentially even related to contamination from non-tumor cells. The comparisons in Supplementary Fig. 6 for example gloss over this important aspect.

We further investigate the hypothesis about cell type specific differences through our single cell RNA sequencing analysis (see major point #2, manuscript section D pages 18-20, lines 277-319). We attempted to evaluate the expression of the observed dysregulated lncRNA and

miRNA, but due to the low sequencing depth of single cell RNA sequencing data the majority of these non-coding RNA were unfortunately not detected (manuscript page 19).

The protein expression comparisons seemed cherry-picked to me. More unbiased analysis there would help support the findings from mRNA data (though presumably the same challenges of controlling for differences in disease type composition and contamination from non-cancer cells would need to be addressed).

Thank you to the reviewer for the feedback. We apologize for the appearance of cherry-picking data. The proteomic comparisons are purposely not unsupervised due to the limited nature of the RPPA data (only 245 probes in the TCPA database and 214 in the MCLP database). Instead, these comparisons are intended to support the unsupervised findings of the gene expression analysis. We have clarified this in the manuscript on page 8, lines 146-148: *“Because the RPPA data includes fewer than 250 proteins, we identified through literature review proteins that are normally highly expressed downstream of each cancer driver pathway, and examined their levels in the RPPA Level 4 Normalized cell line and tumor data (Fig. 2c).”*

4) In general I did not find that the methods were described in sufficient detail. C I found the correlation analysis difficult to interpret. Are the authors arguing that the pattern of expression within these particular pathways is specifically divergent between cell lines and tumors? Assuming they were computing correlation coefficients between cell line and tumor samples using only genes from these particular pathways, this is likely complicated by disease-specific differences in how variably expressed the genes in those pathways are. For instance, in some disease types there might be less variation in expression across genes within a given pathway, and hence lower correlations between any pair of samples (even within tumors). We have added more detailed descriptions of the methods on page 27, lines 486-489: *“All heatmaps use a random subset of samples from each dataset with equal numbers from each disease. The random subset method is used because the CCLE data holds approximately 1/10 the number of samples as the TCGA data, making it impossible to visually detect gene expression differences in heatmaps containing all samples from both datasets.”*

and on page 29, lines 533-536:

“In Python (v3.6.8), the sklearn module (v0.21.2) was used with linear kernel to train a support vector machine (SVM) on 50 random 80/20 splits of the merged TCGA-CCLE gene expression dataset, in which 50 different training sets (and corresponding test sets) were resampled from the same data. The top 10% of genes from each training run based on their feature weights coefficients were merged in a non-duplicate manner, to account for variation in prediction results based on the training and corresponding test set, resulting in 1858 genes.”

We have removed the correlation analysis as we believe it did not add to the fundamental conclusions of the study.

Minor points

1) It wasn't clear to me why the authors chose to focus on the 'KRAS Signaling Up' pathway per se. The argument in section B that differences in overall expression were not a disease-specific artifact was not convincing.

Thank you to the reviewer for identifying an area where clarification is needed. Because the "KRAS Signaling Up" and "KRAS Signaling Down" pathways both represent genes affected as a result of *increased* KRAS pathway activity, we chose to focus on the KRAS Signaling Up pathway because genes overexpressed in response to increased KRAS pathway activity are more likely to be targeted therapeutically (it is difficult to target an underexpressed gene). Focusing on the set of genes more likely to be targeted therapeutically ensures the clinical relevance of our results. We have clarified this in the manuscript on page 7, lines 124-128: *"With respect to KRAS signaling, since both pathways are a result of activated KRAS signaling, all subsequent analysis focuses on KRAS Signaling Up, which represents genes upregulated as a result of activated KRAS signaling. We focused on upregulated genes since most therapeutic approaches focus on reducing the activity of targeted genes¹³."*

2) Why show a random subset of samples in the heatmap plots? These plots could include some additional annotations (such as disease type) or structure (such as hierarchical clustering). As it stands they didn't seem very informative.

Thank you to the reviewer for raising an important point. We visualized all the CCLE data plus a randomly selected disease matched cohort of tumor types because the CCLE data has approximately 1/10 the number of samples as the TCGA data. A heatmap showing all samples from both datasets would be uninformative as it would be impossible to see expression differences between the datasets. We have clarified this in the Methods on page 27, lines 486-489:

"All heatmaps use a random subset of samples from each dataset with equal numbers from each disease. The random subset method is used because the CCLE data holds approximately 1/10 the number of samples as the TCGA data, making it impossible to visually detect gene expression differences in heatmaps containing all samples from both datasets."

3) Why use miRNet predictions to predict DE miRNA from lncRNA, rather than just using the miRNA data itself?

The miRNA data from TCGA and CCLE has a very important limitation: differences in sequencing method. The TCGA miRNA data is Illumina sequencing while the CCLE miRNA data is Nanostring probe. This means that these data are not directly comparable and unsupervised computational comparisons are not reliable. We discuss this in the Methods on page 28, lines 494-503.

4) It's odd that all pathways seem to show higher expression in tumors than cell lines. I wonder if this could be an artifact of the relative nature of expression data and the elimination of immune or non-cancer expression data? In other words, the expression data might need to be renormalized by an adjusted library size after removal of the putative immune genes.

Thank you to the reviewer for the suggestions. However, not all pathways show higher expression in tumors than cell lines. We report an example cancer driver pathway (PI3K-ATK-

mTOR) which was not significantly differentially expressed (Supplementary Figure 2, manuscript page 7, lines 131-134).

5) Proteomics data now exists for >300 cell lines and could be used to provide more systematic comparison of protein expression levels (Nusinow et al., Cell 2020).

Thank you to the reviewer for bringing this excellent study and resource to our attention. However, this study contains mass spectrometry proteomics measurements for only the CCLE database, and not the TCGA database. In our study, we leveraged the RPPA measurements for CCLE and TCGA (MCLP and TCPA) because these measurements were all performed at MD Anderson with the same experimental pipeline, making them directly comparable computationally. It would not be appropriate to compare mass spectrometry CCLE data with RPPA TCGA data.

6) Testing for differences in individual miRNAs after ComBat correction seems odd. I believe the only reason that these differences aren't all identically 0 is based on assumptions of the batch correction model (empirical Bayes moderation methods). I wasn't sure how to interpret this.

We performed ComBat batch effect correction because different miRNA sequencing methods were used in the TCGA and CCLE studies, making it impossible to directly compare the miRNA data without correction. Supplementary Figure 5 shows that the distributions of household miRNA were significantly different in each dataset prior to correction, and that ComBat correction effectively adjusted the expression values so that the distributions are comparable, as would be expected for household miRNA. We did not perform direct unsupervised miRNA comparisons, but only analyzed miRNA identified through lncRNA differential expression in the Illumina mRNA-Seq data. We calculated log fold change using the corrected data, after demonstrating robust correction using housekeeping gene controls. We discuss this in the Methods on page 28, lines 494-503.

7) How do their analyses say anything about PDX and organoids "better recapitulating cell type heterogeneity and cellular interactions"?

We have removed the PDX and organoid section.

Reviewers' comments:

Reviewer #1 (Remarks to the Author):

The authors have satisfactorily addressed the concerns I expressed regarding the effects of tumour purity and tissue heterogeneity on their analysis. Indeed, I am pleased to see that the reanalysis of single-cell sequencing data on HCC yielded evidence suggesting that cancer cell lines may be derived from the most malignant subtypes of tumors.

This is an interesting study that highlights an important topic, and provided that the other reviewers feel that their concerns were addressed, I would be happy to see this article accepted.

Reviewer #2 (Remarks to the Author):

I appreciate the authors thoughtful responses to my concerns, and significant effort in conducting additional analyses; however, my fundamental concerns with the manuscript largely still remain. I don't find that the analyses are sufficiently well-controlled and rigorous to support the key conclusions. I still think the relationship to similar analyses by Yu et al. is not sufficiently addressed. The link between lncRNA, miRNA, and transcriptional differences between cell lines and tumors is an interesting and (as far as I'm aware) novel one, but I still don't see that these connections are sufficiently clearly established by the analyses presented.

A key concern of mine is regarding the SVM analysis approach the authors use to characterize the biological pathways that are differentially active in cell lines and tumors that forms the foundation of the paper. This is a challenging question to address analytically as there are multiple confounding factors, including variable tumor purity and different tumor type composition of the datasets. The authors describe the SVM approach as an 'unsupervised feature identification' method that allows for identification of genes that best differentiate cell lines and tumors regardless of whether each gene is more highly expressed in the cell line and tumor group. Importantly, SVM is a supervised method which is trained to classify samples from a labeled set. In this way, it is not fundamentally different from standard differential expression analysis (both are supervised). The authors also use a linear SVM, where the model coefficients can be interpreted directly as a signature or 'direction in gene space' that best separates cell lines and tumors. Thus, I don't understand why they would want to identify differential genes without regard to the sign of the difference, nor is it clear that they actually did that (they said they took the top 10% of genes based on feature weight coefficients, which for a linear SVM should be signed, unless it's just using the magnitude). If the authors identify the top 10% most predictive genes without regard to sign, and then perform GSEA on that set of genes, it's not clear to me what they would expect to find (gene sets that are most dysregulated but not necessarily knowing whether they are more highly expressed in cell lines or tumors?). I suspect I'm not understanding the full details of what they did, though I did try.

To my mind a standard differential expression analysis has significant advantages in terms of flexibility and interpretability over the multivariate SVM approach used here. Critically, it allows for rigorous control of key confounding variables such as tumor type, and purity. The Yu et al. paper did more or less exactly this and they identified pathways that were clearly up- and down-regulated in cell lines compared to tumors for each tumor type. This to me was far more interpretable biologically. Also, the top results from the Yu et al. paper were not identified here (e.g. cell lines have much higher expression of cell cycle related genes). These apparent differences would need to be addressed more carefully.

The authors approach to account for tumor type and purity differences is to restrict analysis to a few tumor types. I found this too ad-hoc (doing this systematically across tumor types would make sense). In the case of tumor purity effects I don't believe this actually controls for the confounding variable either, just differences in average purity across tumor types (sampling two particular types).

The hypothesis linking differential lncRNA, miRNA, and mRNA expression is an interesting an important part of the paper, but I don't think it's established clearly enough. The differences in tumor type composition between cell line and tumor datasets make this hard to interpret. Other

technical issues confound these analyses as well. For instance, using ComBat to correct for batch effect differences in TCGA and CCLE data could be problematic in the presence of tumor type differences (see Nygaard et al.; 2016; 10.1093/biostatistics/kxv027).

The new single-cell RNA-Seq analysis is an interesting idea to address these issues. Unfortunately, I'm not sure how much conclusion one can draw from that analysis. Critically, it seems there were only two cell lines used (and 7 tumor samples from 2 patients). Thus, it's hard to draw clear conclusions at the general level the authors are discussing throughout the rest of the paper (those two cell lines could be from different HCC subtypes than the patient tumors, or have other random differences). Granted, the authors present this as reinforcing evidence of their main hypotheses, but it wasn't clear to me how exactly they conducted these analyses either (they referred to clustering, but not the actual differential expression analysis methodology), so the fact that the top 100 DE genes in that comparison were significantly enriched in the 4 identified pathways may be pretty weak evidence. These large hallmark cancer pathways tend to involve many variably expressed genes and so can cross a threshold of significance for rather non-specific reasons. Showing the degree of concordance in more detail (likely at the gene level) would build more confidence that there is clear supporting evidence, though it is just from a few samples and there's likely to be huge amounts of sample-to-sample variability.

It was also odd to me that the authors removed substantial aspects of their original paper without much explanation (i.e. the section on PDX and organoids).

In the end, I came away not clearly convinced about the core claims, and I worry that this would leave many readers confused as well.

Reviewer #1 (Remarks to the Author):

The authors have satisfactorily addressed the concerns I expressed regarding the effects of tumour purity and tissue heterogeneity on their analysis. Indeed, I am pleased to see that the reanalysis of single-cell sequencing data on HCC yielded evidence suggesting that cancer cell lines may be derived from the most malignant subtypes of tumors.

This is an interesting study that highlights an important topic, and provided that the other reviewers feel that their concerns were addressed, I would be happy to see this article accepted.

We appreciate Reviewer 1's time and thoughtful comments.

Reviewer #2 (Remarks to the Author):

I appreciate the authors' thoughtful responses to my concerns, and significant effort in conducting additional analyses; however, my fundamental concerns with the manuscript largely still remain. I don't find that the analyses are sufficiently well-controlled and rigorous to support the key conclusions. I still think the relationship to similar analyses by Yu et al. is not sufficiently addressed.

We have added additional clarification on the relationship between our study and the results from Yu et al.; please see Point 4.

1. The link between lncRNA, miRNA, and transcriptional differences between cell lines and tumors is an interesting and (as far as I'm aware) novel one, but I still don't see that these connections are sufficiently clearly established by the analyses presented.

To address the reviewer's concerns that this finding is not sufficiently established, we have moved the discussion on the non-coding RNA findings to the end of the paper and we have significantly reduced the language to reflect the fact that further work is required to fully develop this finding.

2. A key concern of mine is regarding the SVM analysis approach the authors use to characterize the biological pathways that are differentially active in cell lines and tumors that forms the foundation of the paper. This is a challenging question to address analytically as there are multiple confounding factors, including variable tumor purity and different tumor type composition of the datasets. The authors describe the SVM approach as an 'unsupervised feature identification' method that allows for identification of genes that best differentiate cell lines and tumors regardless of whether each gene is more highly expressed in the cell line and tumor group. Importantly, SVM is a supervised method which is trained to classify samples from a labeled set. In this way, it is not fundamentally different from standard differential expression analysis (both are supervised). The authors also use a linear SVM, where the model coefficients can be interpreted directly as a signature or 'direction in gene space' that best separates cell

lines and tumors. Thus, I don't understand why they would want to identify differential genes without regard to the sign of the difference, nor is it clear that they actually did that (they said they took the top 10% of genes based on feature weight coefficients, which for a linear SVM should be signed, unless it's just using the magnitude). If the authors identify the top 10% most predictive genes without regard to sign, and then perform GSEA on that set of genes, it's not clear to me what they would expect to find (gene sets that are most dysregulated but not necessarily knowing whether they are more highly expressed in cell lines or tumors?). I suspect I'm not understanding the full details of what they did, though I did try.

We apologize to the reviewer that the Methods were not clear. We have added clarifying language in the Methods:

The top 10% of genes from each training run based on the magnitude of their feature weights coefficients were merged in a non-duplicate manner, to account for variation in prediction results based on the training and corresponding test set, resulting in 1858 genes.

We would also like to refer the reviewer to our explanation of methods and reasoning in Section A, which clarifies these issues:

Feature identification using SVM allowed us to identify the set of genes which best differentiate between tumor and cell line regardless of whether each gene is more highly expressed in the tumor group or the cell line group. The SVM approach has been shown to eliminate noisy results in a high-dimensional gene expression comparison, by drawing out the greatest sources of variation across all samples^{9,10}.

3. To my mind a standard differential expression analysis has significant advantages in terms of flexibility and interpretability over the multivariate SVM approach used here. Critically, it allows for rigorous control of key confounding variables such as tumor type, and purity. The Yu et al. paper did more or less exactly this and they identified pathways that were clearly up- and down-regulated in cell lines compared to tumors for each tumor type. This to me was far more interpretable biologically.

Classical differential expression (DE) analysis has many strengths and has been used for knowledge gain in many studies, including Yu et al. At the request of the reviewer, we performed classical DE analysis on our data and saw comparable results to the SVM analysis. However, we chose to use the SVM approach because classical DE also has known limitations¹. Importantly, statistical methods such as DE rely on assumptions about RNAseq data and its distribution, and the significance of differentially expressed genes can be confounded by sample size and other experimental factors². These methods are also subject to the multiple testing problem, because each gene is treated as a separate testing hypothesis², and these issues are further exacerbated in situations where multiple tissues or diseases are involved in the same dataset¹.

In our study, we wished to build upon previous work by conducting a different yet complementary analysis. Machine learning analysis such as SVM does not rely on assumptions about the data; the only important factor is which genes are the most useful for discriminating between groups based on their expression¹. We chose to leverage such a generalizable method

in order to develop inferences about important genes which differentiate *tumor* from *cell culture* in a complex dataset with many factors. This approach allows us to derive widely applicable knowledge about the differences that arise in gene expression when cancer cells are grown in cell culture.

4. Also, the top results from the Yu et al. paper were not identified here (e.g. cell lines have much higher expression of cell cycle related genes). These apparent differences would need to be addressed more carefully.

We would like to clarify that we did in fact recapitulate the top results of the Yu et al. paper (please refer to Supplementary Table 1). We identified several developmental, immune, and cell cycle related pathways in our analysis. However, in our study, we decided to explore further the cancer driver pathways, since they are most likely to be targeted for treatment in these cell lines. We have added clarifying language in Section A as follows:

We categorized the pathways into 6 categories (cellular response, development, cancer driver, metabolism, blood, and immune). Notably, our results recapitulate the findings of a recent study by Yu and colleagues which found key differences in developmental, cell cycle, and immune pathways between cell lines and tumors (Supplementary Table 1, tab 3). In this study, we were particularly interested in the 5 cancer driver pathways which overlapped between the coding genes GSEA and the lncRNA-derived GSEA, as these molecular pathways are most likely to be the focus of preclinical trials in cancer cell lines.

5. The authors approach to account for tumor type and purity differences is to restrict analysis to a few tumor types. I found this too ad-hoc (doing this systematically across tumor types would make sense). In the case of tumor purity effects I don't believe this actually controls for the confounding variable either, just differences in average purity across tumor types (sampling two particular types).

In order to account for tumor purity differences, we performed the SVM analysis using only the data from the tumor types with the highest (KIRC) and lowest (PRAD) overall tumor purity. We also addressed the tumor purity issue with our single cell RNAseq analysis (Section C), which provides further detail to our results by demonstrating that enrichment of the KRAS and IL2/STAT5 pathways appear to be partly dependent on normal cell infiltrate.

6. The hypothesis linking differential lncRNA, miRNA, and mRNA expression is an interesting and important part of the paper, but I don't think it's established clearly enough. The differences in tumor type composition between cell line and tumor datasets make this hard to interpret.

Please see point 1.

7. Other technical issues confound these analyses as well. For instance, using ComBat to correct for batch effect differences in TCGA and CCLE data could be problematic in the presence of tumor type differences (see Nygaard et al.; 2016; 10.1093/biostatistics/kxv027).

A limitation of any analysis using these datasets is that some of these data come from different platforms. We avoided batch effect correction whenever possible, but for the miRNA data it was necessary to correct for batch effects due to the different platforms.

8. The new single-cell RNA-Seq analysis is an interesting idea to address these issues. Unfortunately, I'm not sure how much conclusion one can draw from that analysis. Critically, it seems there were only two cell lines used (and 7 tumor samples from 2 patients). Thus, it's hard to draw clear conclusions at the general level the authors are discussing throughout the rest of the paper (those two cell lines could be from different HCC subtypes than the patient tumors, or have other random differences). Granted, the authors present this as reinforcing evidence of their main hypotheses, but it wasn't clear to me how exactly they conducted these analyses either (they referred to clustering, but not the actual differential expression analysis methodology), so the fact that the top 100 DE genes in that comparison were significantly enriched in the 4 identified pathways may be pretty weak evidence. These large hallmark cancer pathways tend to involve many variably expressed genes and so can cross a threshold of significance for rather non-specific reasons. Showing the degree of concordance in more detail (likely at the gene level) would build more confidence that there is clear supporting evidence, though it is just from a few samples and there's likely to be huge amounts of sample-to-sample variability.

In order to address the reviewer's concerns on sample-to-sample variability and heterogeneity, we have performed 2 additional single cell RNA sequencing analysis using 2 sets of matched brain tumor and cell lines from the same patients (see the end of Section C, and Supplementary Figures 8 and 9):

A limitation of the HCC analysis is that the cancer and cell line samples were not matched from the same patients. Therefore, we performed the same analysis using two sets of matched brain tumor and cell line samples (melanoma brain metastases; Supplementary Figures 8 and 9 respectively). We again observe enrichment of the same four cancer driver pathways, with the exception of the KRAS Signaling Pathway in the first analysis.

9. It was also odd to me that the authors removed substantial aspects of their original paper without much explanation (i.e. the section on PDX and organoids). In the end, I came away not clearly convinced about the core claims, and I worry that this would leave many readers confused as well.

We agreed with the reviewer that the section on PDX and organoids was not developed enough, and so we removed this section in order to make the paper more focused.

References:

1. Bzdok, D., Altman, N. & Krzywinski, M. Statistics versus machine learning. *Nat. Methods* **15**, 233–234 (2018).
2. Love, M. I., Huber, W. & Anders, S. Moderated estimation of fold change and dispersion for

RNA-seq data with DESeq2. *Genome Biol.* **15**, 550 (2014).

Reviewers' comments:

Reviewer #2 (Remarks to the Author):

In the latest revisions of their manuscript, the authors have addressed some of my specific concerns, however, several of my fundamental concerns largely remain. To my mind, the analysis strategy lacks robustness, the analysis itself is insufficiently rigorous in places, and the interpretation of results is unsupported by the data at times. Additionally, I believe there is insufficient detail and clarity in the presentation of the results, and the methods description is not clear and detailed enough to enable others to reproduce the analysis.

More specific points:

SVM methodology:

I now feel I understand the analyses the authors have done to identify differentially expressed pathways, and I think it's a generally reasonable approach, however certain aspects remain ad-hoc and unjustified, and overall I find the authors' conclusions are overly reliant on this SVM-GSEA approach.

-The authors report that they found 'compatible' results using standard DE analysis, and also that their results were compatible with the DE analysis of Yu et al. Directly showing these comparisons could aid in the interpretation of their results and better put them in context of closely related work. For example, the lack of gene sets enriched in the cell lines compared to tumor samples (particularly cell cycle pathways such as E2F_TARGETS and G2M_CHECKPOINT) seems a clear difference from the Yu et al results that would warrant some explanation.

- As the authors don't report anything about the accuracy of their SVM classifier (which I do think would be of interest), I don't understand why they have used the train/test splitting procedure. One reason could be to obtain uncertainty estimates on the feature importance estimates themselves (via bootstrap resampling), however, the authors do not report such information (or even the actual feature importance values). It seems it would be significantly more straightforward to simply train the model on all the data and report a single set of feature importance values.

- The justification provided for using SVM over standard DE analysis is a bit questionable in my mind. I wouldn't argue that standard hypothesis testing approaches are without their issues, but it seems that the authors have circumvented challenges with quantifying statistical significance by using ad-hoc strategies (i.e. take top 10% most important features from any training set).

- The authors' conclusions rely very heavily on GSEA with the HALLMARK gene set collections, which are very large gene sets, and inevitably contain some bias. I think the result could be strengthened and clarified by showing gene-level results, and by using alternative gene set collections (e.g. KEGG, Reactome, GO).

-Why not run the SVM analysis on each individual cancer type and show those results, as they do in a few examples, and as done in Yu et al? I understand the focus is on pan-cancer signatures, but this would add additional clarity to the sources of difference and how they vary across cancer types, and would help further control for potential bias due to differences in cancer type compositions between tumor and cell line datasets.

-The authors could also include purity estimates as a covariate in their SVM to better control for purity bias

Results and Methods Reporting

-The authors should make their analysis code available to aid reproducibility and also clarify details of the analysis

- The authors should report the gene-level statistics (e.g. signed model coefficients) in the supplementary tables

- The authors should provide GSEA stats in all tables (rather than lists of gene sets that meet their significance criteria)

-Sometimes 'significantly' enriched gene sets are based on p-values, other times it's not clear. In general, q-values should be used when testing across all gene sets.

-There are a number of places where figure contents are unclear. For example, in the heatmaps it's not clear what 'relative expression' represents exactly. Are these log-TPM values? It seems like these values are different in the bulk and single-cell analyses (Fig 2b and 3d), but it's not clearly stated.

-There remain several places where the analysis details are not clearly presented. For example, I found it difficult to follow exactly how their scRNA-Seq DE analysis was done. The authors mention using clustering. I assume this was done to classify cancer and normal cell populations in the tumor samples. Also, they say the "Wilcoxon signed-rank" test was used to identify differentially expressed genes. Perhaps they mean the "rank-sum" test, as the signed-rank test is a one-sample or paired test which I don't think would be applicable in this case.

Interpretation of Results:

- The authors seem to move between suggesting differences between tumors and cell lines are due to selection bias (i.e. certain types of tumors are more amenable to in vitro growth), and that the differences result from within-sample changes induced by in vitro culture conditions, and deploy these interpretations in somewhat confusing ways. For example they state "Interestingly, all 4 pathways show much higher overall expression in tumors than in cell lines, indicating that these genes are downregulated as a result of the transition from tumor to cell culture dish (Fig. 2b)." However such a difference could also be due to selection bias, which is an interpretation they seem to favor elsewhere in the text.

-The authors claim that Fig S1 shows that the immune related pathways are unrelated to expression of cancer driver genes, however Fig S1b seems to show some driver genes that are significantly correlated with expression of immune pathways. Showing that the distribution of pairwise correlations has a median value near zero does not seem a convincing argument that immune (or more broadly any normal cell) contamination won't bias their results

- The observation that there is much overlap in the GSEA results of the protein-coding genes, and the overlap of the protein coding and miRNA-targets genes is expected at least to some degree by construction. A more informative test would be to assess the overlap of the full set of miRNA-target genes (without first intersecting with the differentially expressed protein-coding genes)

- Fig 2d: it's unclear what conclusions are being drawn from this figure, and it's also not referenced in the main text.

-It's not clear from Fig 3b that "the 4 cancer driver pathways are significantly enriched in the HCC tumor cells". In particular I don't see IL2/STAT5 signaling in the list in Fig 3b or 3c. Also, it's not clear that KRAS_SIGNALING_UP is enriched in the tumor samples in 3b.

-The cell type classification in Fig S8 and S9 looks questionable (particularly in S9 where some of the clusters labeled as tumor cells appear closer to the astrocyte or microglia clusters). If this analysis is to be included I think some additional validation of the cell type classification would be needed.

Reviewer #2 (Remarks to the Author):

In the latest revisions of their manuscript, the authors have addressed some of my specific concerns, however, several of my fundamental concerns largely remain. To my mind, the analysis strategy lacks robustness, the analysis itself is insufficiently rigorous in places, and the interpretation of results is unsupported by the data at times. Additionally, I believe there is insufficient detail and clarity in the presentation of the results, and the methods description is not clear and detailed enough to enable others to reproduce the analysis.

We thank the reviewer for taking the time to review our manuscript and provide feedback.

More specific points:

SVM methodology:

I now feel I understand the analyses the authors have done to identify differentially expressed pathways, and I think it's a generally reasonable approach, however certain aspects remain ad-hoc and unjustified, and overall I find the authors' conclusions are overly reliant on this SVM-GSEA approach.

-The authors report that they found 'compatible' results using standard DE analysis, and also that their results were compatible with the DE analysis of Yu et al. Directly showing these comparisons could aid in the interpretation of their results and better put them in context of closely related work. For example, the lack of gene sets enriched in the cell lines compared to tumor samples (particularly cell cycle pathways such as E2F_TARGETS and G2M_CHECKPOINT) seems a clear difference from the Yu et al results that would warrant some explanation.

We have a different approach, so we expect the exact gene set results to be different. These gene sets are representations of biological activity. As we noted previously, our results do show cell cycle activity differences between cell line and tumor such as "HALLMARK_MITOTIC_SPINDLE".

- As the authors don't report anything about the accuracy of their SVM classifier (which I do think would be of interest), I don't understand why they have used the train/test splitting procedure. One reason could be to obtain uncertainty estimates on the feature importance estimates themselves (via bootstrap resampling), however, the authors do not report such information (or even the actual feature importance values). It seems it would be significantly more straightforward to simply train the model on all the data and report a single set of feature importance values.

In machine learning, the feature importances are derived from training on a training set, and then predicting on a test set, and calculating which features were the most important for predictions. The suggested procedure is not appropriate for a machine learning approach.

- The justification provided for using SVM over standard DE analysis is a bit questionable in my mind. I wouldn't argue that standard hypothesis testing approaches are without their issues, but it seems that the authors have circumvented challenges with quantifying statistical significance by using ad-hoc strategies (i.e. take top 10% most important features from any training set).

A recent paper on an automated machine learning platform "JADBio" is extremely illustrative to answer this question. <https://www.biorxiv.org/content/10.1101/2020.05.04.075747v1.full>

"Non-linear models (Random Forests, non-linear SVMs, and Decision Trees) are selected as best in 88% of the time, suggesting that **modern analysis of biological data should not be limited to standard, linear, statistical models.**"

Analysis of multi-omics cancer data using SVM found that "just 2 out of the 22796 measured biomarkers carry all information to almost perfectly classify the 4 sub-types of TET [Thymic Epithelial Tumors]". ... "The first such set of two biomarkers identified (called the reference signature) is the expression values of the gene CD3E and the miRNA miR498, respectively (see Supplementary Methods for markers' names codebook). miR498 is the marker most associated (pairwise) to the outcome. However, CD3E ranks only 190 in terms of pairwise association with the outcome (p-values 10e-94 and 10e-56, respectively)! **In other words, if one performs standard differential expression analysis, they will have to select 189 other markers before reaching CD3E.** In contrast, JADBIO's feature selection algorithms, **recognize and filter out the redundant features.** This example anecdotally illustrates that the most predictive signature is not always composed of the most differentially expressed quantities, but rather by predictors who complement each other in terms of informational content. It is actually possible that markers with no pairwise association to (not differentially expressed by) the outcome to be necessary for optimal prediction²⁹. These examples illustrate the difference between standard differential expression analysis and signature discovery for predictive modeling."

In other words: classification with methods like SVM can identify genes most important for differentiating between 2 groups and can filter out redundant features (extremely common in correlative molecular data such as gene expression).

Therefore, our approach to take the top 10% of features is extremely conservative due to the correlative and redundant nature of gene expression data. In order to be even more conservative, we implemented the 50x bootstrapping approach to ensure that we are not achieving biased results from a single classification.

- The authors' conclusions rely very heavily on GSEA with the HALLMARK gene set collections, which are very large gene sets, and inevitably contain some bias. I think the result could be strengthened and clarified by showing gene-level results, and by using alternative gene set collections (e.g. KEGG, Reactome, GO).

The HALLMARK gene set collection has only 50 gene sets (each with 100 or 200 genes per gene set) whereas KEGG has 186 gene sets, Reactome has 1615 gene sets, and GO has 10,402 gene sets. HALLMARK is one of the most specific and focused gene set collections which summarizes well defined biological gene sets.

-Why not run the SVM analysis on each individual cancer type and show those results, as they do in a few examples, and as done in Yu et al? I understand the focus is on pan-cancer signatures, but this would add additional clarity to the sources of difference and how they vary across cancer types, and would help further control for potential bias due to differences in cancer type compositions between tumor and cell line datasets.

This is beyond the scope of this study.

-The authors could also include purity estimates as a covariate in their SVM to better control for purity bias

There is no such thing as a covariate in an SVM.

Results and Methods Reporting

-The authors should make their analysis code available to aid reproducibility and also clarify details of the analysis

We will make our final publication code publicly available on GitHub at <https://github.com/lauren-sanders/TCGA-CCLE>

- The authors should report the gene-level statistics (e.g. signed model coefficients) in the supplementary tables. The authors should provide GSEA stats in all tables (rather than lists of gene sets that meet their significance criteria)

These stats have been added to all tables that did not previously have them.

-Sometimes 'significantly' enriched gene sets are based on p-values, other times it's not clear. In general, q-values should be used when testing across all gene sets.

We use p-value calculated from FDR q-value in all cases, as shown in the Supplementary Tables.

-There are a number of places where figure contents are unclear. For example, in the heatmaps it's not clear what 'relative expression' represents exactly. Are these log-TPM values? It seems like these values are different in the bulk and single-cell analyses (Fig 2b and 3d), but it's not clearly stated.

For bulk RNAseq, heatmaps are generated using seaborn clustermap method in Python with z-scores calculated per gene. "Relative expression" indicates the values that are plotted after z-score calculation. For single cell RNAseq, heatmaps are generated using scanpy heatmap

method in Python with values scaled between 0 and 1. These are standard methods for both types of data. We have clarified this in the Methods section.

-There remain several places where the analysis details are not clearly presented. For example, I found it difficult to follow exactly how their scRNA-Seq DE analysis was done. The authors mention using clustering. I assume this was done to classify cancer and normal cell populations in the tumor samples. Also, they say the "Wilcoxon signed-rank" test was used to identify differentially expressed genes. Perhaps they mean the "rank-sum" test, as the signed-rank test is a one-sample or paired test which I don't think would be applicable in this case.

The Wilcoxon rank-sum test was used to identify differentially expressed genes between groups of single cells. We have clarified this and the rest of the scRNAseq methods in the manuscript.

Interpretation of Results:

- The authors seem to move between suggesting differences between tumors and cell lines are due to selection bias (i.e. certain types of tumors are more amenable to in vitro growth), and that the differences result from within-sample changes induced by in vitro culture conditions, and deploy these interpretations in somewhat confusing ways. For example they state "Interestingly, all 4 pathways show much higher overall expression in tumors than in cell lines, indicating that these genes are downregulated as a result of the transition from tumor to cell culture dish (Fig. 2b)." However such a difference could also be due to selection bias, which is an interpretation they seem to favor elsewhere in the text.

We have clarified the interpretations in the text. In particular, the sentence above has now been changed to: "Interestingly, all 4 pathways show much higher overall expression in tumors than in cell lines (Fig. 2b)." in order to avoid conjecture.

-The authors claim that Fig S1 shows that the immune related pathways are unrelated to expression of cancer driver genes, however Fig S1b seems to show some driver genes that are significantly correlated with expression of immune pathways. Showing that the distribution of pairwise correlations has a median value near zero does not seem a convincing argument that immune (or more broadly any normal cell) contamination won't bias their results

Fig S1 shows that in general, there is limited relationship between the cancer driver genes and immune gene expression. However, we allow for the possibility that normal cell contamination could bias our results, and we explore this possibility in our multiple single cell RNAseq analyses that we have added. The final takeaway from our paper is that gene expression from cell lines is used for hypothesis generation for drug targets and we demonstrate differences between cell lines and tumors that should be taken into account in these situations.

- The observation that there is much overlap in the GSEA results of the protein-coding genes, and the overlap of the protein coding and miRNA-targets genes is expected at least to some degree by construction. A more informative test would be to assess the overlap of the full set of

miRNA-target genes (without first intersecting with the differentially expressed protein-coding genes)

This is not a correct description of the analysis that we performed. The text from the manuscript is below:

*To investigate potential non-coding RNA dysregulation in cell lines as compared to tumors, we focused on the **54 lncRNA identified as differentially expressed through the SVM classification** (Fig. 1, Supplementary Table 1, tab 1). We used miRNet databases to **link the 54 differentially expressed lncRNA to the 4 differentially expressed cancer driver pathways via shared miRNA interactions** (Supplementary Table 2).*

Via miRNet, we found that 77 miRNA have known interactions both with genes in the four cancer driver pathways, and with 11 of the differentially expressed lncRNA (11 lncRNA: LBX2-AS1, CERS6-AS1, DLGAP1-AS1, H19, IQCH-AS1, LINC00240, LINC00665, LINC00707, LINC00847, LINC00622, LIMD1-AS1). With the exception of LINC00707, 10 of the 11 lncRNA are significantly underexpressed in CCLE (Fig. 4b,c). We hypothesized that the reduced cell line expression of these lncRNA may be associated with expression changes in the downstream miRNA regulatory network, which in turn may be associated with aberrant expression of the four cancer driver pathways being controlled by the miRNA network.

In order to investigate this hypothesis, we leveraged publicly available miRNA sequencing (miRNAseq) data from CCLE and TCGA. We used the ComBat method to correct for experimental batch effects (see Methods, Supplementary Fig. 5)32. Sixty-nine of the 77 miRNA were quantified in both miRNAseq datasets, so we used these miRNA for all downstream analyses (Supplementary Table 4, tab 1). We calculated the log fold change (LFC) in expression between CCLE and TCGA for these 69 miRNA. Notably, over half of the miRNA (n=43) are more highly expressed in cell lines than tumors.

- Fig 2d: it's unclear what conclusions are being drawn from this figure, and it's also not referenced in the main text.

We have clarified this in the text on page 8: "These four pathways overall have similar correlation between tumor and cell line samples when evaluated on a disease-specific basis (Fig. 2d)."

-It's not clear from Fig 3b that "the 4 cancer driver pathways are significantly enriched in the HCC tumor cells". In particular I don't see IL2/STAT5 signaling in the list in Fig 3b or 3c. Also, it's not clear that KRAS_SIGNALING_UP is enriched in the tumor samples in 3b.

We have clarified this item in the manuscript on page 11-12: "GSEA of the top 100 genes identified the KRAS, TP53, TNFA via NFKB, and IL3/STAT3 pathways as enriched in genes

differentially expressed in HCC tumor cells (pvalue < 0.05), very similar to the bulk RNA sequencing analysis (Fig. 3b).”

-The cell type classification in Fig S8 and S9 looks questionable (particularly in S9 where some of the clusters labeled as tumor cells appear closer to the astrocyte or microglia clusters). If this analysis is to be included I think some additional validation of the cell type classification would be needed.

We have provided validation of the cell type classification by providing cluster plots colored by the marker genes we used for classification. As is evident from the additional gene cluster plots we provided, there are clear gene expression differences between the tumor clusters in SFig9 and the microglia and astrocyte clusters (e.g. CD68, a known marker of microglia, is not expressed at all in the nearby tumor cluster).